https://doi.org/10.1038/s41467-019-13551-1　　**OPEN**

# A primate-specific retroviral enhancer wires the *XACT* lncRNA into the core pluripotency network in humans

Miguel Casanova [1,2]*, Madeleine Moscatelli[1,2], Louis Édouard Chauvière[1], Christophe Huret[1], Julia Samson[1], Tharvesh Moideen Liyakat Ali[1], Olga Rosspopoff[1] & Claire Rougeulle [1]*

Transposable elements (TEs) have been proposed to play an important role in driving the expansion of gene regulatory networks during mammalian evolution, notably by contributing to the evolution and function of long non-coding RNAs (lncRNAs). *XACT* is a primate-specific TE-derived lncRNA that coats active X chromosomes in pluripotent cells and may contribute to species-specific regulation of X-chromosome inactivation. Here we explore how different families of TEs have contributed to shaping the *XACT* locus and coupling its expression to pluripotency. Through a combination of sequence analysis across primates, transcriptional interference, and genome editing, we identify a critical enhancer for the regulation of the *XACT* locus that evolved from an ancestral group of mammalian endogenous retroviruses (ERVs), prior to the emergence of *XACT*. This ERV was hijacked by younger hominoid-specific ERVs that gave rise to the promoter of *XACT*, thus wiring its expression to the pluripotency network. This work illustrates how retroviral-derived sequences may intervene in species-specific regulatory pathways.

[1] Université de Paris, Epigenetics and Cell Fate, CNRS, F-75013 Paris, France. [2] These authors contributed equally: Miguel Casanova, Madeleine Moscatelli. *email: miguel.casanova@univ-paris-diderot.fr; claire.rougeulle@univ-paris-diderot.fr

More than half of the human genome is composed of transposable elements (TEs), which are mobile genomic DNA elements capable of autonomous and non-autonomous replication[1,2]. The vast majority of human TEs are retrotransposons, which can be divided into three classes: long interspersed elements, short interspersed elements, and endogenous retroviruses (ERVs). Classically seen as junk and parasitic DNA, the last decades have brought TEs into the limelight as important drivers of genome evolution[3–6]. TEs have been a major force for the evolution of genomes, promoting structural variation, genome size expansion, tridimensional organization, genetic diversity, as well as affecting gene regulation[4,7–9]. Their recognized role as a source of regulatory innovation stems from the co-option of TEs by their hosts, either creating or modifying genes (both coding and non-coding), or by acting as regulatory elements[10–12]. Indeed, TE-derived regulatory elements, such as the long terminal repeats (LTRs) flanking ERVs, are abundantly distributed across the human genome and have the ability to rewire genes into coordinated networks of transcriptional regulation. For example, certain classes of ERVs, such as HERVK and HERVH, have been shown to influence the expression of genes in pluripotent contexts[13–17]. This occurs through the emergence of novel genes containing the promoter sequence of the ERVs or by rewiring the expression of existing genes located in the vicinity of an ERV-derived enhancer. In both cases, the ERVs provide transcription factor-binding platforms for master regulators of pluripotency, such as OCT4, SOX2, and NANOG, which wire the expression of these genes to pluripotency.

However, the uncontrolled transposition of TEs in the genome can potentially have detrimental effects for its host. Thus, several mechanisms exist that keep TEs in check to prevent deleterious changes in the host genome[18]. The delicate balance resulting from this skirmish between the host and TEs has to ensure that essential biological processes are conserved, whereas, at the same time, allowing the emergence of novel regulatory mechanisms. For this reason, the presence of TEs in exonic sequences of protein-coding genes is rather low, likely as a result of counter-selection. In contrast, TEs are over-represented in vertebrate long non-coding RNAs (lncRNAs), with almost all of them containing at least one TE in their sequence[19–21]. This led to the proposal that TEs form basic sequence and structural blocks in lncRNAs, which collectively influence their function[22]. This alliance between TEs and lncRNAs constitutes an intricate and rich strategy to control biological processes that are tightly linked to the developmental state. A paradigm of such a process is X-chromosome inactivation (XCI). In females of all mammalian species, XCI achieves transcriptional shutdown of one X chromosome, by converting it into facultative heterochromatin. XCI takes place during early development and is generally established at the exit of pluripotency[23]. Its initiation is under the control of a region on the X chromosome called the X-inactivation center (XIC) and chromosome silencing is triggered by the accumulation of a lncRNA produced from the XIC, XIST[24,25]. Besides XIST, several other lncRNA genes are found within the XIC and have been shown to play diverse roles in the regulation of XCI[26]. Interestingly, all of the lncRNAs found within this region have evolved from the pseudogenization of protein-coding genes driven by the integration of different TEs[27–29].

In human, XIST starts being expressed from the eight-cell stage, concomitantly with zygotic genome activation, and from all X chromosomes, including in males[30–32]. Whereas the accurate timing of human XCI has not yet been firmly documented[33,34], in these early stages of pre-implantation development there is a transient uncoupling between the expression of XIST and XCI[33,34]. This raises the question as to how X chromosomes are mechanistically protected from being silenced in the initial stages when XIST starts being expressed and how is XCI coupled to a later developmental stage in humans. We have previously identified XACT, a repeat-rich lncRNA, which has the striking property of coating active X chromosomes in early human embryonic stages[35]. Studies in human embryonic stem cells (hESCs) and human embryos suggest that XACT can affect XIST expression, localization, or activity in these contexts[34,36]. Thus, XACT could act as a transient XIST antagonist, ensuring that XCI is established at the right developmental stage. Understanding how this lncRNA evolved in humans and the mechanisms linking its expression to pluripotent contexts is thus of the uttermost importance.

In this study, we explore the contribution of distinct classes of ERVs in the molecular coupling of XACT expression to pluripotency. Through an analysis of the XACT surrounding region across primates and using a combination of transcriptional interference and genome-editing approaches in hESCs, we identify a critical genomic element required for XACT expression. We show that this element, which acts as an enhancer, belongs to a family of ERVs found across mammalian species. Our findings suggest an exaptation of an ancient ERV by younger hominoid-specific ERVs that gave rise to XACT and illustrate how retroviral-derived sequences may intervene in species-specific regulatory pathways.

## Results

**ERV elements drove the emergence of XACT and T113.3.** To explore the emergence and regulation of XACT, we revisited the organization and evolution of this locus. The XACT gene is located in a large intergenic region on the X chromosome between the protein-coding genes AMOT and HTR2C[35]. Another gene, T113.3, is found ~50 kb upstream of XACT and has been previously characterized as giving rise to a spliced and cytoplasmic transcript[35]. Transcript assembly reconstruction using Scallop[37] and complementary DNA cloning and sequencing of RNA from hESCs revealed that the T113.3 transcript consists of three exons (Supplementary Fig. 1A). Using CPAT[38] we revealed that this transcript has a low coding potential and likely acts as a lncRNA (Supplementary Fig. 1A). Whereas the T113.3 gene is predicted to have a functional potential[39], its function is still unknown.

We analyzed the organization of this region in humans in comparison with five other primate species (chimpanzee, gorilla, gibbon, rhesus macaque, and marmoset) and observed an overall conservation of the syntenic region extending from the LHFPL1 to the LRCH2 genes (upstream of AMOT and downstream of HTR2C, respectively) (Fig. 1a). All protein-coding genes (LHFPL1, AMOT, HTR2C, and LRCH2) display the same genomic organization, orientation, and high exon sequence identity throughout all primate species. In contrast, XACT and T113.3 show a limited sequence identity across primates, particularly in species more distantly related to humans (Fig. 1a). Notably, the sequences corresponding to the promoter region of XACT and T113.3 are conserved in hominoids, but not in rhesus macaque or more distant primate species (Fig. 1b). This suggests that the emergence of these two genes is a recent evolutionary event that occurred concomitantly in the genome of the last common ancestor of macaque and gibbons some 20 Myr ago (Fig. 1c).

In hESCs, XACT is expressed from the minus strand, whereas T113.3 is expressed from the opposite strand (Fig. 1d). In hESCs, the transcription start sites (TSS) of XACT and T113.3 are enriched in H3 Lys4 trimethylation (H3K4me3), suggesting these are the bona-fide promoters of their respective genes (Fig. 1d). In induced pluripotent stem cells (iPSCs) from chimpanzee, XACT is similarly expressed, but not T113.3, despite its sequence conservation (Supplementary Fig. 1B). As expected, no expression is detected from these loci in macaque, due to the lack of the

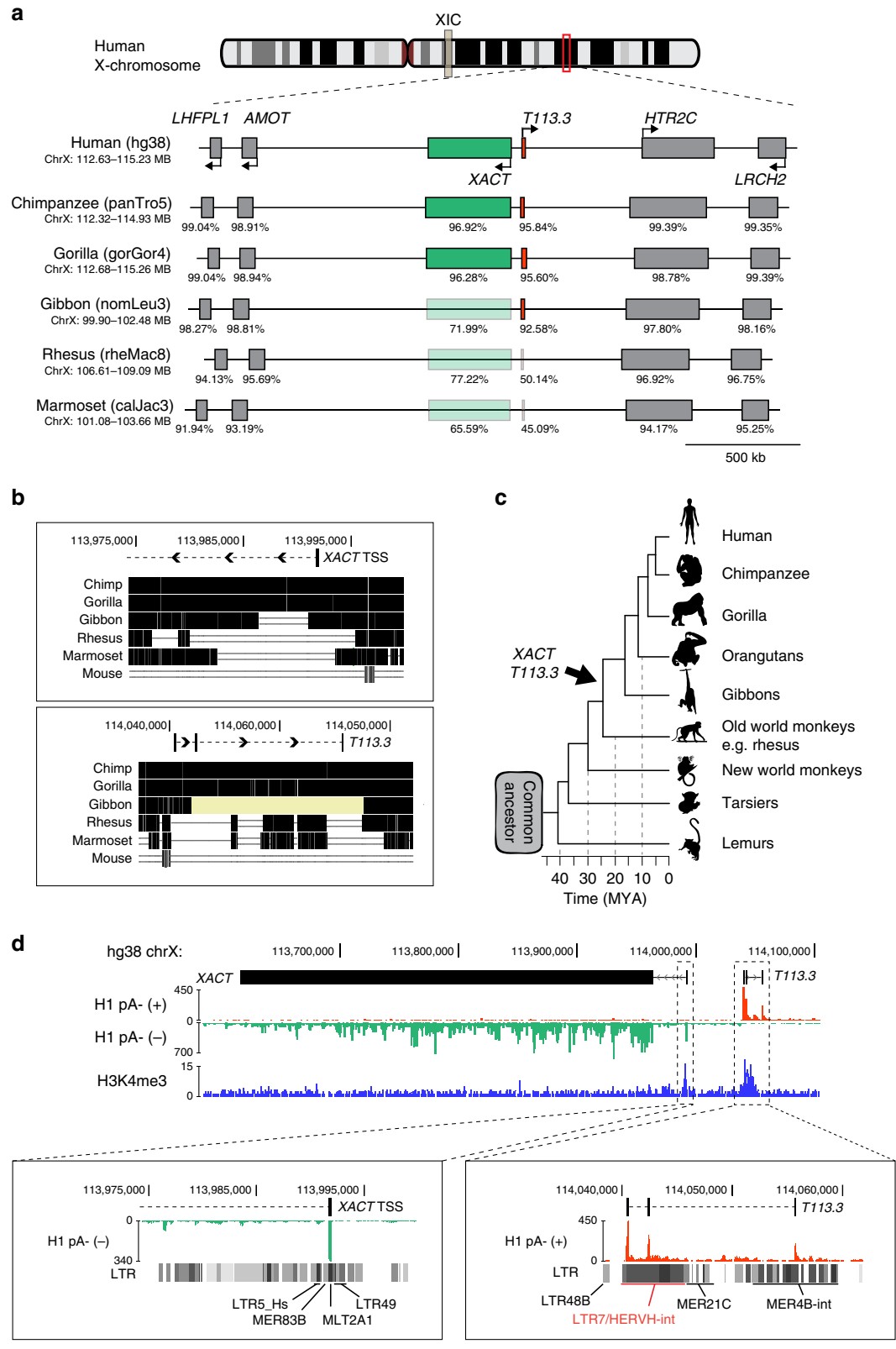

promoters for both *XACT* and *T113.3* (Fig. 1b and Supplementary Fig. 1B). Analysis of the repeat composition of *XACT* and *T113.3* promoters revealed that both were shaped by the insertion of several LTR/HERV elements. The region surrounding the promoter of *XACT* is composed of fragments of LTRs belonging to different groups: LTR5_HS, MER83B, MLT2A1, and LTR49. Rapid amplification of cDNA ends has previously shown that the MLT2A1 element corresponds to the TSS of *XACT*[35]. The *T113.3* gene derives from the insertion of a 5.1 kb fragment of a HERVH element, including its 5′-LTR7 promoter, which acts as the TSS

**Fig. 1 *XACT* and *T113.3* derive from different classes of ERVs. a** Map of the syntenic genomic region, from *LHFPL1* to *LRCH2* genes, in different primate species. Sequences of all human genes from the locus were extracted and compared with the orthologous sequences in primates, using blast[59]. Sequence identity was performed using MAFFT multiple alignment tool with default parameters[61]. Percentage of sequence identity is represented under each gene on the locus, for the different species (cDNA sequence identity for protein-coding genes and DNA sequence identity for *XACT* and *T113.3* genes). **b** Multiple alignment across six species (other hominoids: chimpanzee, gorilla, and gibbon; rhesus macaque, lemur, and mouse) for the human *XACT* promoter and *T113.3* gene. The multiple alignment was performed using Multiz online from the UCSC browser. Positions on the X chr are indicated for genome assembly hg38. **c** Phylogenetic tree of primates, with evolutionary distance between species. The approximate evolutionary time for the appearance of the LTR/ERVs giving rise to the promoters of *XACT* and *T113.3* are represented. **d** ENCODE strand-specific RNA-seq from male H1 hESCs[57] showing transcription over a large region encompassing the *XACT* and *T113.3* genes from the minus and plus strand, respectively. ChIP-seq data for H1 hESCs from the ENCODE project[57] showing distribution of H3K4me3 along the *XACT*/*T113.3* locus, with observable peaks over the TSS of both genes. A zoom of the TSS of *XACT* and *T113.3* shows the different classes of LTRs that define the promoter of both genes.

for *T113.3*. In addition, other parts of the GAG- and POL-coding regions, as well as other HERVs, such as MER21C and MER4B, constitute the remaining of the gene (Fig. 1d). Altogether, this indicates that *XACT* and *T113.3* share a similar evolutionary history, which is mostly dependent on the introduction of hominoid-specific ERVs in the *XACT*/*T113.3* loci.

**Correlated expression dynamics of *XACT* and *T113.3*.** To understand whether the transcription of *XACT* and *T113.3* is controlled by a shared regulatory network, we first explored the patterns and dynamics of their expression in hESCs and upon their differentiation. We observed, using RNA-fluorescence in situ hybridization (FISH), that *XACT* and *T113.3* are co-expressed from the active X chromosome (Xa) in both male (H1) and female (H9) hESCs before differentiation, with a minority of cells expressing either transcript individually (Fig. 2a and Supplementary Fig. 2A). Their expression dynamics is also strongly correlated across a 10-day differentiation time course of male and female hESCs (Fig. 2b and Supplementary Fig. 2B), with both genes being turned off from day 3 onwards.

We further explored the transcriptional dynamics of *XACT* and *T113.3* in human pre-implantation embryos using public single-cell RNA sequencing (RNA-seq) datasets[33,40–42]. Expression from both loci can be detected concomitantly between the four- and the eight-cell stages, likely following the major wave of zygotic gene activation (Supplementary Fig. 2C). Furthermore, the expression detected from the *XACT* and *T113.3* loci is correlated at early (E3 to early E5) and late (E5 to E7) stages of development in both female and male embryos (Fig. 2c). A weaker transcriptional correlation between *XIST* and *T113.3* is observed in early stages of pre-implantation development, but is mostly lost at later stages (Supplementary Fig. 2D). These observations confirm that the *XACT* and *T113.3* loci display similar transcriptional dynamics in different pluripotent contexts, in hESCs and in early embryos, suggesting that they may be co-regulated.

To corroborate this hypothesis, we analyzed Hi-C datasets[43] from H1 hESCs and from human foreskin fibroblasts (HFF), in which *XACT* and *T113.3* are expressed and silenced, respectively. In hESCs, the 5′-region of *XACT* is embedded within a local compartment that includes *T113.3* and expands into the upstream *LRCH2* gene, whereas in fibroblasts this local compartmentalization is absent (Fig. 2d). This indicates that the regulatory sequences of *XACT* and *T113.3* belong to the same TAD in cells where both genes are expressed.

**Blocking *T113.3* transcription impairs *XACT* expression.** To test whether the expression of *XACT* or of *T113.3* are inter-dependent, we performed CRISPR interference (CRISPRi), using a catalytically dead Cas9 coupled with a repressive KRAB domain (dCas9-KRAB), to induce local heterochromatinization of the TSS regions of *XACT* and *T113.3*[44]. Given the similarity of *XACT*/*T113.3* expression dynamics in female and male

pluripotent contexts, we performed all our functional assays in male H1 hESCs. We first generated H1 hESCs stably expressing the CRISPRi machinery together with gRNAs targeting the promoter of either *XACT* or *T113.3* (Fig. 3a, top scheme and Supplementary Fig. 3A). Reverse transcription–quantitative PCR (RT-qPCR) analysis showed the efficiency of the CRISPRi system, with the levels of *XACT* and *T113.3* RNAs being strongly reduced when their respective promoter is targeted (Fig. 3a). Although interfering with *XACT* expression does not affect the expression levels of *T113.3*, targeting the *T113.3* promoter induces a significant decrease in *XACT* lncRNA levels (Fig. 3a). As targeted cells maintain a normal morphology and stable expression of pluripotency markers (Supplementary Fig. 3B), this effect is unlikely due to an alteration of the pluripotent state that could have resulted from the transcriptional interference of *T113.3*. Moreover, accumulation of the H3K9me3 mark induced by *T113.3* CRISPRi spreads only locally, up to 6 kb away from the targeted site, but does not reach the *XACT* promoter (Fig. 3b), whereas the distribution of H3K4me3 and H3K27ac changes only slightly across the loci (Supplementary Fig. 3C). Altogether, this suggests that the *T113.3* locus participates in the regulation of *XACT* expression, either through the *T113.3* transcript, the act of transcription, or the chromatin landscape around the LTR7 forming the *T113.3* promoter.

**T113.3 is dispensable for *XACT* expression.** The LTR7/HERVH class of retrotransposons has been linked to the transcriptional control of human pluripotency[15,17]. We thus hypothesized that the LTR7/HERVH defining the *T113.3* gene could link the expression of the *XACT*/*T113.3* locus to the pluripotency network. To test this hypothesis, we developed parallel strategies to independently assess the involvement of the *T113.3* RNA or of the *T113.3* locus in the regulation of *XACT*.

First, we knocked down (KD) the *T113.3* RNA using LNA-gapmers (locked nucleic acid, antisense oligonucleotides) targeting different intronic or exonic regions (Fig. 4a, upper panel). As a control, we designed LNA-gapmers for different regions of the *XACT* transcript. H1 hESCs transfected with *XACT*- or *T113.3*-specific LNA-gapmers show a significant reduction of their respective target (Fig. 4a), without any morphological alteration or changes in the expression of pluripotency or differentiation markers (Supplementary Fig. 4A). However, interfering with the levels of either *T113.3* or *XACT* RNAs does not lead to any changes on the RNA levels of the other transcript (Fig. 4a), indicating that the *T113.3* transcript is dispensable for the expression of *XACT* in hESCs (and vice versa).

We next tested the role of the *T113.3* LTR7/HERVH in the regulation of *XACT*, by using CRISPR/Cas9 to delete a 25 kb region encompassing the entire *T113.3* gene (including the whole LTR7/HERVH element at its 5′-end) in H1 hESCs (Supplementary Fig. 4bB). We screened individual colonies by PCR for the presence of a deletion or an inversion of the *T113.3* gene and

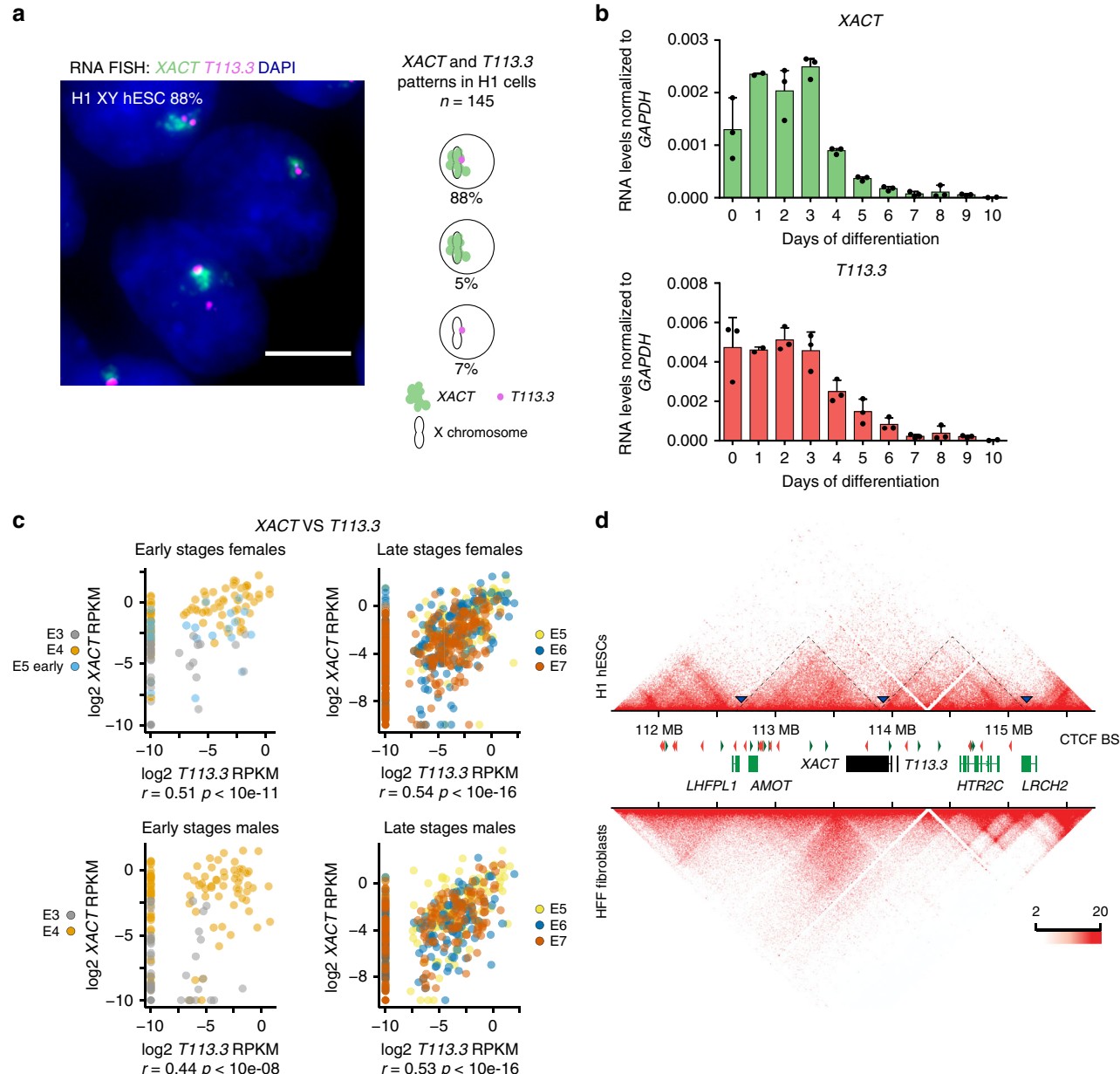

**Fig. 2 XACT and T113 are co-regulated in pluripotent contexts. a** Analysis of *XACT* (green) and *T113.3* (magenta) expression by RNA-FISH in male H1 hESCs. Percentage of cells co-expressing *XACT* and *T113.3* from the same locus is indicated. Quantification of the different patterns of expression is represented on the right (*n* = 145). The white scale bar represents 5 μm. **b** RT-qPCR analysis of *XACT* and *T113.3* expression during a 10-day undirected differentiation of H1 hESCs. The bar charts correspond to the average of three independent differentiation experiments. Error bars indicate the standard deviation (SD). The correlation coefficient for the expression dynamics of the two genes is shown (*r* = 0.94, Pearson correlation). **c** Plot of *T113.3* vs. *XACT* expression levels (log2 reads per kilobase per million mapped reads [RPKM]) in early (E3, E4, and early E5) and late stage (E5, E6, and E7) female (upper panels) and male (lower panels) pre-implantation embryos, with corresponding Spearman's correlation score and *p*-value. Dataset from ref. [33]. **d** Hi-C heatmap of the *LHFPL1/LRCH2* locus in H1 hESCs and in HFF [43] visualized with Juicebox[70]. Red and green arrows indicate the orientation of CTCF binding sites. Blue arrows and dotted lines show the boundaries of the domains in H1 hESCs.

selected three independent clones of each genotype (wild type (WT), deleted, and inverted; Supplementary Fig. 4B) for further analysis. All clones, independently of their genotype, showed comparable expression levels of X-linked (*AMOT*, *HTR2C*, *ATRX*), pluripotency (*OCT4*), and differentiation marker genes (*NODAL*) (Supplementary Fig. 4C). As expected, *T113.3* RNA could not be detected in clones carrying a deletion of the locus; in addition, the inversion of the *T113.3* gene lead to the complete abrogation of *T113.3* transcription (Fig. 4b). However, none of these mutations affected the expression levels of *XACT* in hESCs

(Fig. 4b). In addition, the expression dynamics of *XACT* during undirected differentiation of the *T113.3* mutant hESCs (Fig. 4c) and the global kinetics of the differentiation were unaffected (Supplementary Fig. 4D). These results clearly show that the LTR7/HERVH defining the *T113.3* gene is dispensable for the expression of *XACT* in hESCs and for its silencing dynamics during cellular differentiation.

**Identification of a pluripotent-specific enhancer.** The analysis of H3K27ac distribution across the locus, a mark of active

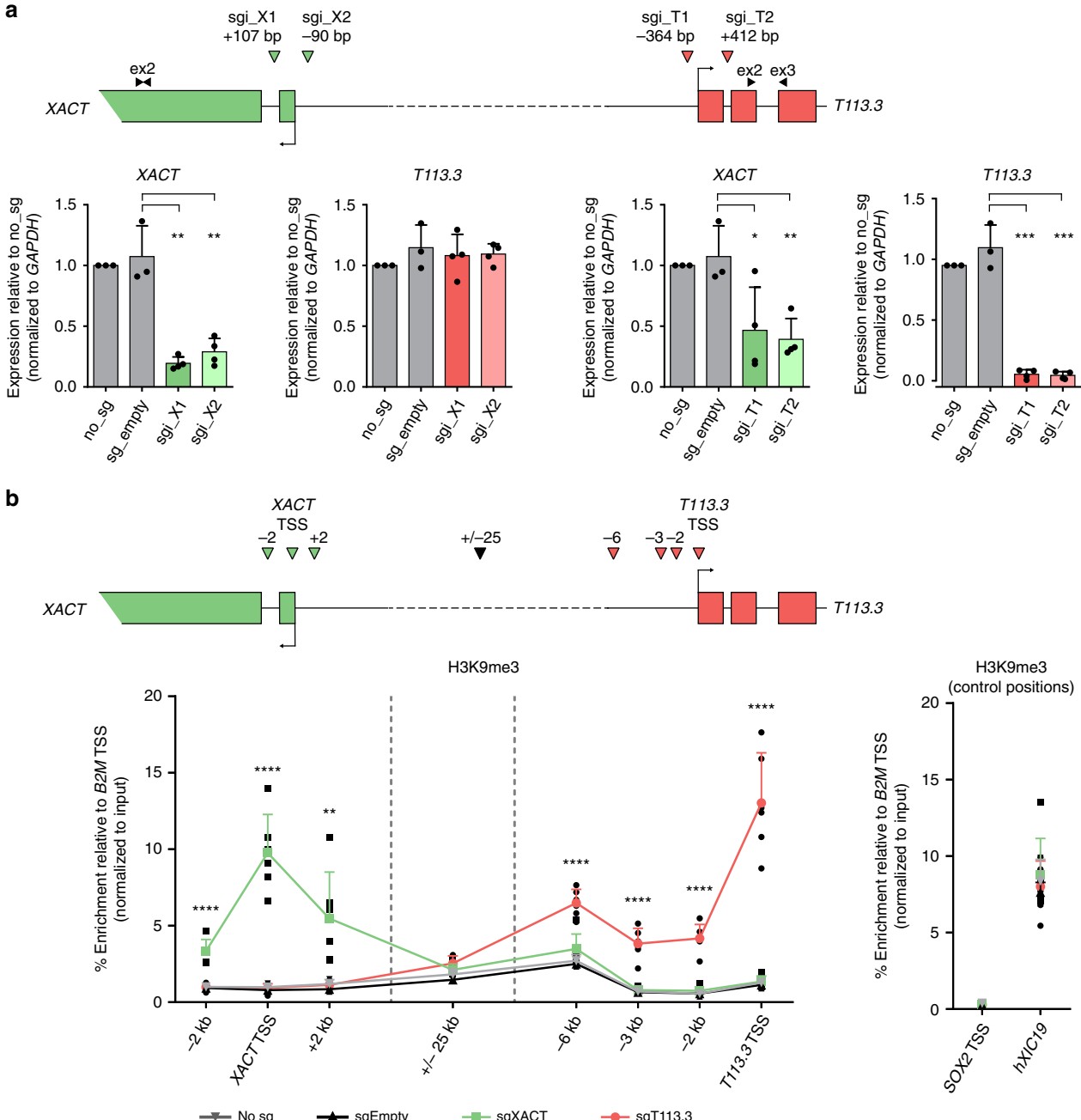

**Fig. 3 Transcription of *T113.3* regulates the expression of *XACT* in hESCs. a** The top scheme depicts the CRISPRi strategy used to target the *XACT* and *T113.3* promoters, with the positions of the two sgRNAs (colored arrows) and primer pairs used to measure the transcript levels of each gene (black arrows). Quantification of the global RNA levels of *XACT* and *T113.3* by RT-qPCR upon *XACT* CRISPRi (left bar charts) and *T113.3* CRISPRi (right bar charts). Error bars represent the SD of at least four different passages of hESCs infected with guides targeting either *XACT* or *T113.3*. **b** ChIP-qPCR analysis of H3K9me3 enrichment across the *XACT/T113.3* locus upon *XACT* CRISPRi (green squares) or *T113.3* CRISPRi (red circles) in hESCs. The positions analyzed are indicated in the scheme with arrows. ChIP-qPCR data for untransfected cells (gray inverted triangles) and cells transfected with a vector without a sgRNA (black triangles) is also shown. Right graph represents the relative enrichment of H3K9me3 in control positions (*SOX2* TSS and hXIC19). Error bars represent the SD of three passages of each of two independent hESC lines infected with different guides targeting either *XACT* or *T113.3* ($n = 6$). *$P$-values < 0.05, **$P$-values < 0.01, ***$P$-values < 0.001, and ****$P$-values < 0.0001. Statistical significances were determined using a one-way ANOVA with sg_empty.

enhancers, pointed to a potential regulatory region approximately 2 kb upstream of the *T113.3* TSS (Supplementary Fig. 3C). Examination of ATAC-seq and chromatin immunoprecipitation sequencing (ChIP-seq) datasets from H1 hESCs[45] confirmed the existence of a broad acetylation domain, which is not found in fetal fibroblasts (Fig. 5a and Supplementary Fig. 5A), suggesting that this region may serve as a pluripotent-specific enhancer.

Further inspection of ChIP-seq profiles revealed that this domain encompasses a CTCF peak and binding sites for several transcription factors, including pluripotency factors OCT4, SOX2, and NANOG.

To test the enhancer activity of this domain and identify which element(s) are involved in the regulation of *XACT* and *T113.3*, we used CRISPR/Cas9 to delete either the CTCF or the transcription

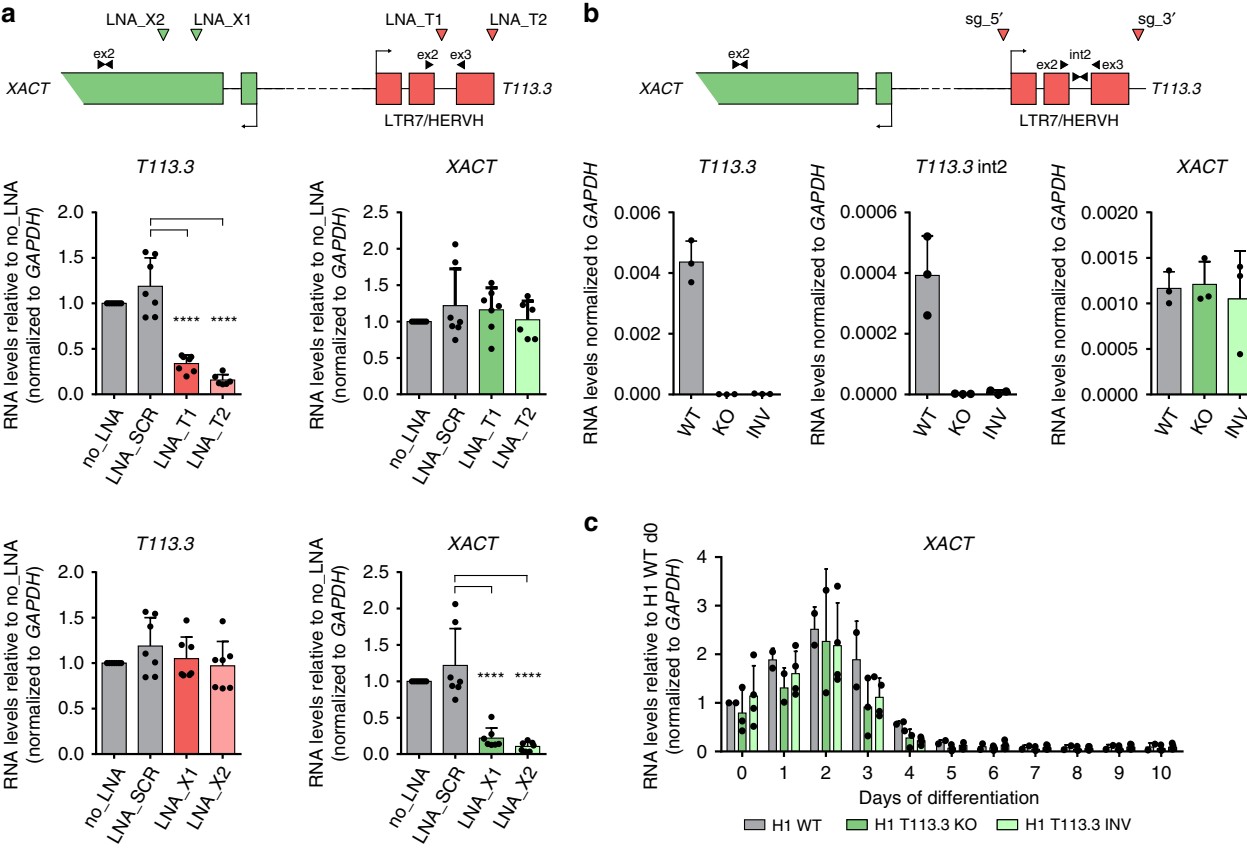

**Fig. 4 Neither *T113.3* gene nor *T113.3* RNA regulate *XACT* expression. a** Scheme of *XACT* and *T113.3* loci showing the position of the LNA GapmeRs used for *XACT* (green arrows) and *T113.3* (red arrows) KD, and the position of the primers used for the RT-qPCR (black arrows). RT-qPCR analysis of *T113.3* and *XACT* expression in cells transfected with LNA GapmeRs targeting *T113.3* (top panels) or *XACT* (bottom panels) (minimum of six replicates per condition). **b** Scheme of the *XACT* and *T113.3* locus showing the guide positions (red arrows) used for the deletion of the *T113.3* gene and the position of the primers used for RT-qPCR (black arrows). RT-qPCR analysis of *T113.3* and *XACT* expression in H1 WT, *T113.3* KO, and INV hESC clones. n = 3. **c** RT-qPCR analysis of *XACT* expression during a 10-day undirected differentiation of H1 T113.3 WT, KO, and INV hESC lines. n ≥ 3. Error bars represent the SD. Statistical significances were determined using a one-way ANOVA (all tests compared to LNA_SCR). *P-values < 0.05, **P-values < 0.01, ***P-values < 0.001, and ****P-values < 0.0001.

factor-binding (TFB) sites (Supplementary Fig. 5B). We selected at least two independent clones of non-mutated, deleted, or inverted genotypes (Supplementary Fig. 5B). All these clones retain normal expression levels of pluripotent markers (*NANOG*, OCT4), indicating that none of these mutations significantly impact pluripotency (Supplementary Fig. 5C). Deletion or inversion of the CTCF site did not affect steady-state levels of *T113.3* or *XACT* transcripts compared with WT H1 ESCs (Fig. 5b). In contrast, deletion of the TFB site completely abolished *T113.3* transcription and drastically reduced the levels of *XACT* RNA (Fig. 5b). Interestingly, inversion of the TFB site did not perturb the expression of neither *XACT* nor *T113.3*, further supporting the role of this genomic element as an enhancer, acting independently of its orientation. Thus, we conclude that a pluripotent-specific proximal enhancer, upstream of the *T113.3* gene, is essential for the regulation of the *XACT*/*T113.3* loci in hESCs.

To confirm the role of the pluripotency transcription factors OCT4, SOX2, and NANOG in the activity of this enhancer, we used small interfering RNAs (siRNAs) to KD their expression, either individually or in combination. Western blotting analysis revealed that the simultaneous targeting of *SOX2* and *OCT4* resulted in the depletion of all three master regulators of the pluripotency network (Fig. 5c), consistent with the fact that OCT4 positively regulates NANOG[46]. In this condition specifically, we observed a significant reduction of *T113.3* and *XACT* transcript levels (Fig. 5d and

Supplementary Fig. 5D), further supporting the dependence of the locus towards master regulators of the pluripotency circuitry. We excluded an indirect effect linked to KD-induced differentiation, as upregulation of differentiation markers is observed in all conditions where OCT4 is targeted, whereas the effect on the *XACT*/*T113.3* loci is seen only when the three factors are simultaneously depleted (Supplementary Fig. 5D). Although we cannot exclude that these transcription factors play a more complex role regulating *XACT* and *T113.3*, namely by directly acting at their TSS, the combined results of the siRNA KD and LTR48B deletion suggests that the enhancer upstream of *T113.3* connects the transcriptional regulation of the locus to the pluripotency network in hESCs.

**Evolutionary history of the *XACT* enhancer.** A closer analysis of the *XACT*/*T113.3* enhancer revealed that it derives from an ancient LTR48B from the ERV1 family (Fig. 6a). Unlike *XACT* and *T113.3*, which originate from the insertion and exaptation of hominoid-specific LTRs, the LTR48B enhancer is found in all primate species, suggesting that it was introduced in a common ancestor of primates, at least 20 Myr before the LTR/HERV insertions that gave rise to both *XACT* and *T113.3* promoters (Fig. 6b). We scanned distinct position weight matrices (PWMs) for core pluripotency transcription factors against the human X-linked LTR48B enhancer element and identified potential binding sites for OCT4, SOX2, and NANOG (Fig. 6c). These binding sites are conserved in all primate species

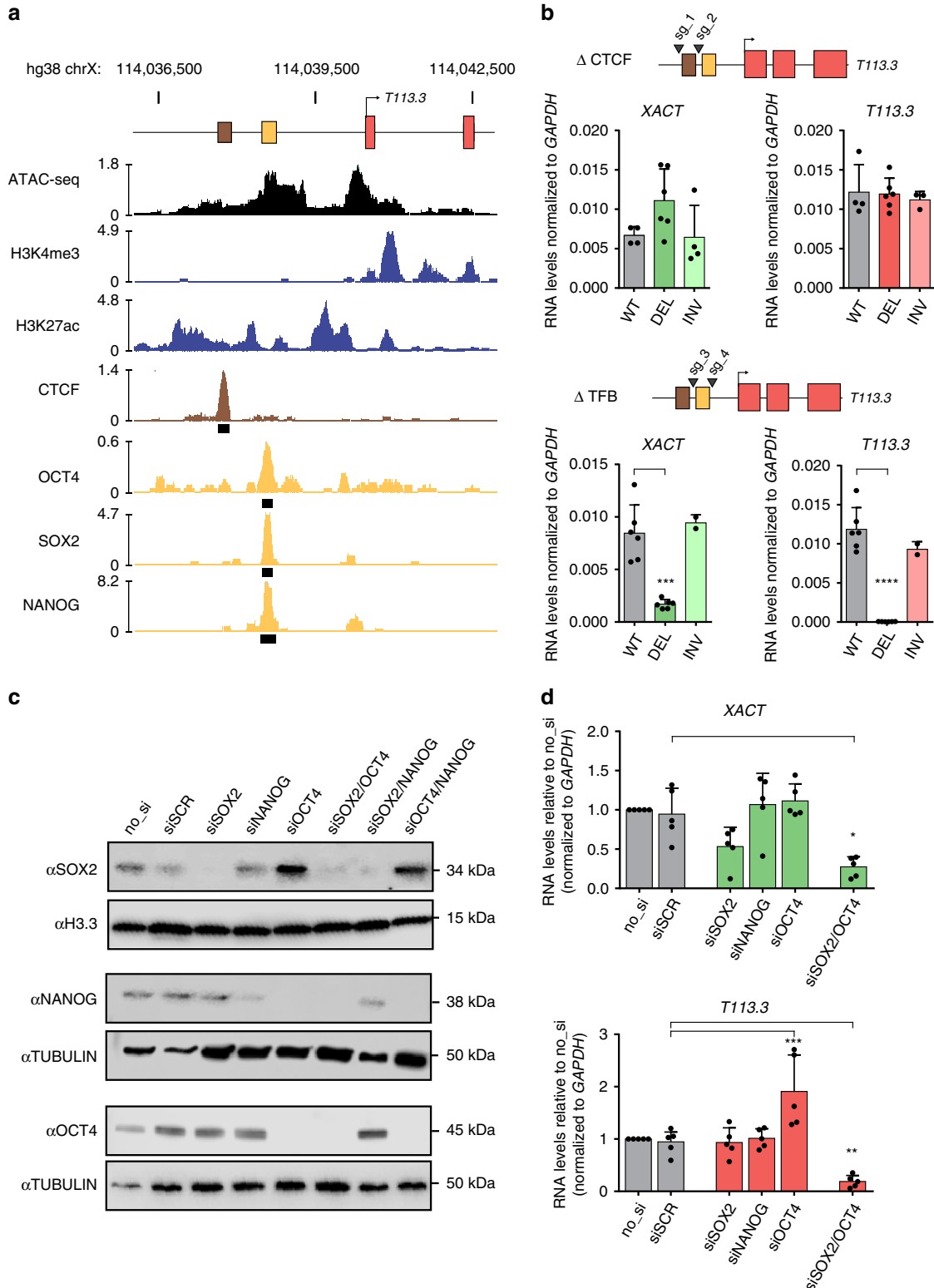

analyzed here, suggesting that this LTR element evolved as a primate-specific enhancer, the activity of which is connected to the pluripotency machinery. To gain further insight into the evolution of the activity of this enhancer in the primate lineage, we explored available iPSC RNA-seq and ChIP-seq datasets from humans and rhesus macaques[47]. As expected, in iPSCs of Rhesus, whose genome lacks the LTR elements defining the promoters of *XACT* and *T113.3*,

no expression of either gene is observed (Supplementary Fig. 6A). Nonetheless, the LTR48B element coincides with a peak of H3K27ac (Supplementary Fig. 6A), suggesting that this element harbors an active enhancer in Rhesus iPSCs, despite the absence of transcription at the syntenic positions of the *XACT* and *T113.3* genes.

Information about the evolution of the LTR48B family in primates is scarce. We thus investigated whether this family of

**Fig. 5 A common enhancer wires the _XACT/T113.3_ locus to the pluripotency network. a** ATAC-seq (black) and ChIP-seq data for H3K4me3, H3K27ac (blue), CTCF (brown), OCT4, SOX2, and NANOG (orange) enrichment over the _T113.3_ TSS region in H1 hESCs. Publicly available data were obtained from the ENCODE project[57] and CISTROME DB[45]. Called peaks for CTCF, OCT4, SOX2, and NANOG are shown as black rectangles. **b** Scheme of the _T113.3_ locus showing the guide positions (black arrows) used for targeting either the CTCF (top panel) or TFB (bottom panel) sites upstream of the _T113.3_ gene. Quantitative RT-PCR analysis of _XACT_ (left) and _T113.3_ (right) expression in CTCF- (top panel) or TFB- (bottom panel) targeted hESCs. The bar charts correspond to the average of at least two independent clones. **c** KD efficiency of _SOX2_, _OCT4_, and _NANOG_ using an siRNA approach was assessed by western blotting. H1 hESCs were used as a non-transfected control cell line (no_si). siSCR was used as a non-targeting siRNA. Tubulin and H3.3 were used as loading control. **d** Quantification of _XACT_ (top bar chart) and _T113.3_ (bottom bar chart) steady-state RNA levels by RT-qPCR in cells transfected with siRNAs targeting SOX2, OCT4, NANOG, or a combination of siRNAs targeting both OCT4 and SOX2. Error bars indicate the SD. Statistical significances were determined using a one-way ANOVA (all conditions compared with WT). *$P$-values < 0.05, **$P$-values < 0.01, ***$P$-values < 0.001, and ****$P$-values < 0.0001.

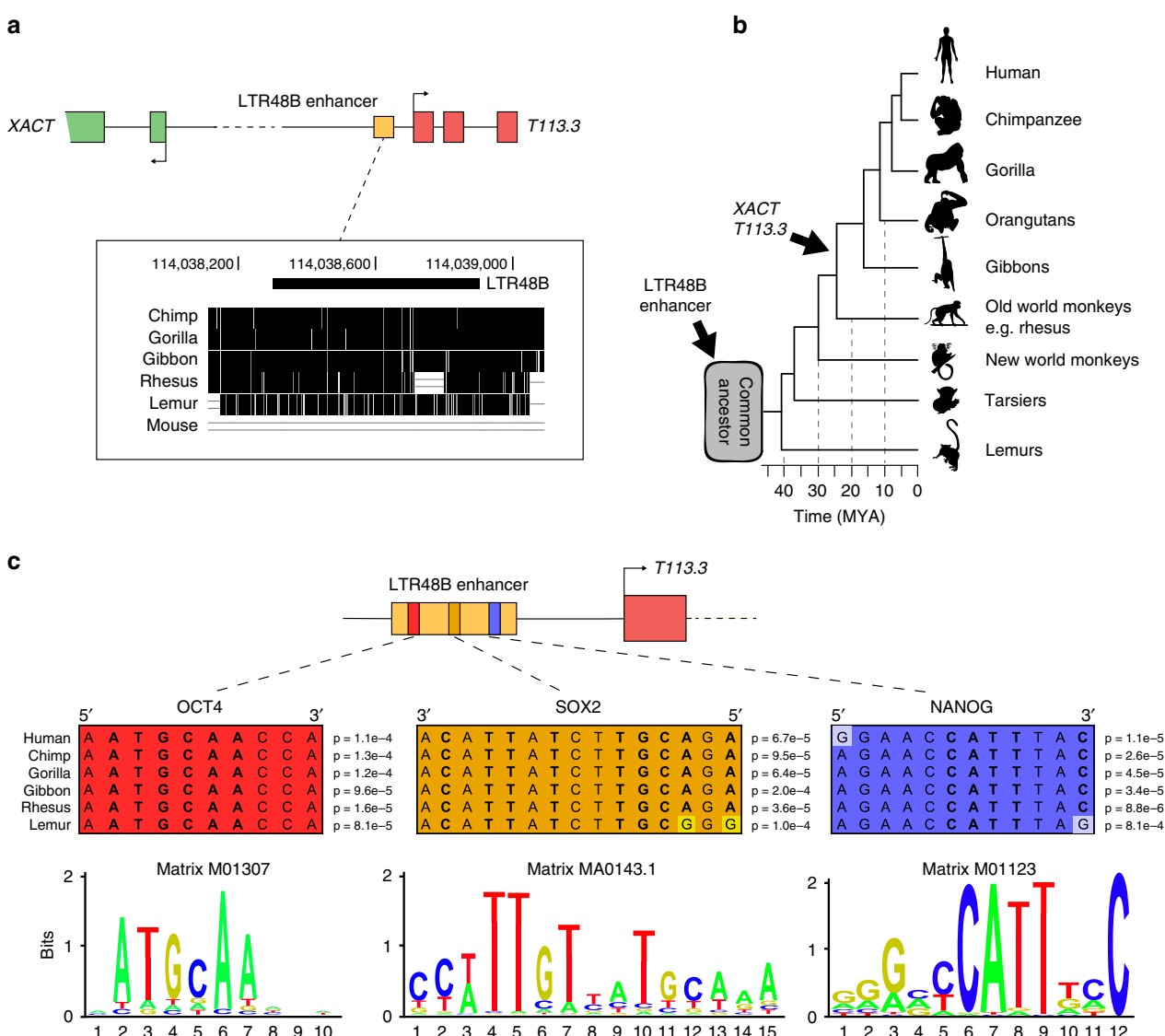

**Fig. 6 Evolution of the LTR48B-derived enhancer. a** Multiple alignment across six species (three hominoids: chimpanzee, gorilla, gibbon; rhesus macaque, lemur, and mouse) for the human LTR48B element defining the common _XACT_ and _T113.3_ enhancer. The multiple alignment was performed using Multiz online from the UCSC browser. Positions on the X chr. are indicated for genome assembly hg38. **b** Phylogenetic tree of primates, with evolutionary distance between species. The approximate evolutionary time for the appearance of the LTR48B enhancer and the LTR/ERVs giving rise to the promoters of _XACT_ and _T113.3_ are represented. **c** Prediction of binding sites for pluripotent transcription factors OCT4, SOX2, and NANOG in the LTR48B enhancer. Using RSAT matrix-scan[62] and position-specific scoring matrices (PSSMs) of pluripotent transcription factors as input, the most probable binding sites were identified in the human LTR48B _XACT/T113.3_ enhancer. The PSSMs used to scan the region of interest were the Transfac M01307 for OCT4, M01123 for NANOG, and the Jaspar MA0143.1 for SOX2. Multiple alignment for the putative binding sites was performed on orthologous sequences in chimpanzee, gorilla, gibbon, rhesus, and lemur. _P_-values for the likelihood of the identified regions being bona-fide binding sites in the different primate species, were determined using RSAT matrix-scan. Logos of the PSSMs used for each transcription factor are represented below.

retroelements may have been co-opted into the pluripotency network of host cells. We scanned the consensus sequence of human LTR48B for the presence of putative binding sites for core pluripotency transcription factors (Supplementary Fig. 6B). Strikingly, the consensus sequence of the LTR48B group lacks potential binding sites for OCT4, SOX2, or NANOG. Accordingly, the majority of LTR48B elements found in the human genome are not bound by OCT4 and NANOG in available ChIP-seq datasets from hESCs[48] (Supplementary Fig. 6C,D). Out of the 1026 LTR48B found in the human genome, only 34 are bound by NANOG, 8 of which are also bound by OCT4 (Supplementary Fig. 6E). This is in contrast with LTR7-HERVH elements, which are bound by OCT4 and NANOG genomewide (Supplementary Fig. 6C–E), in agreement with this HERV family being essential for the maintenance of pluripotency in human cells[15,17]. Collectively, our data show that an ancient LTR48B element located between *XACT* and *T113.3* has evolved independently of the bulk of the LTR48B family, acquiring binding sites for core pluripotency factors. Recruitment of these factors created an active enhancer in pluripotent cells, which was in turn co-opted by younger LTRs inserted in the locus, giving rise to two novel genes *XACT* and *T113.3*.

## Discussion

Recent studies have demonstrated the importance of families of TEs in nuclear organization and gene expression regulation[49–51]. These crucial studies provide general rules about families of TEs and their role, but lack formal demonstration of their regulatory function as well as finer details about the mechanisms used by individual TEs to regulate specific loci in the genome (discussed in ref. [3]). Indeed, a recent study demonstrating that only a minority of TEs with enhancer features has a role in the regulation of gene expression calls for a critical assessment of the functional role of individual TEs[52,53]. Here we explored the evolutionary history of the *XACT* locus to decipher the molecular coupling of *XACT* expression to pluripotency in hESCs. Our study reveals that both *XACT* and an upstream gene, *T113.3*, originate from the insertion of elements of various TE classes in the hominoid lineage. In particular, genomic remnants from ancient ERV insertions gave rise to the promoters that drove the emergence of these genes. Moreover, we explore how the expression of these TE-derived genes are wired into the pluripotency transcriptional network. We show that *XACT* and *T113.3* have hijacked an ancestral LTR fragment that acts as an enhancer in pluripotent contexts in all primate species, coupling their expression to early embryogenesis.

It was previously shown that different families of ERVs are dynamically expressed, in a stage- and cell-specific manner, during early development[54]. Importantly, families of hominoid-specific ERVs, such as LTR7/HERVH and LTR5_Hs/HERVK, were shown to rewire different genes into the pluripotent transcriptional network, in part by creating binding platforms for combinations of pluripotency-associated transcription factors[17,51,55]. This led us to hypothesize that the *T113.3* gene, a highly expressed LTR7/HERVH element located upstream of *XACT*, could couple its expression to the pluripotent context. However, we demonstrate that neither the LTR7/HERVH *T113.3* gene nor its transcript influence the transcription of *XACT* in this cellular context. Although the importance of the LTR7/HERVH family of retroelements in pluripotency seems undeniable, our work highlights how a careful analysis of specific loci are required to ascertain the regulatory role of individual TEs. Further investigation will be necessary to understand the role of the *T113.3* gene and transcript (if any) in XCI or in pluripotency.

Although the younger LTR7/HERVH element defining the *T113.3* gene does not function as an enhancer for *XACT*, we identified another LTR-derived fragment that takes on this role. This enhancer belongs to the LTR48B subfamily, a more ancient group of ERV1 retroelements that is found across mammalian species[11]. The *XACT*/*T113.3* enhancer was inserted more than 40 Myr ago and is found across all primate species, where it shows features of an active enhancer, even in species in which *XACT* and *T113.3* are not present, such as rhesus macaque. Importantly, this element seems to have evolved independently from the core of the LTR48B subfamily and has the ability to act as a binding platform for pluripotency transcription factors, such as OCT4, SOX2, and NANOG. This appears to be a very rare event, as only 34 insertions among the 1026 present in the human genome are bound by at least 1 of these TFs. This suggests that the binding sites were either lost in the core of the LTR48B subfamily or, more likely, acquired in a subset of LTR48B elements. It would be interesting to explore this subset of LTR48B elements and test their influence on the transcriptional status of surrounding genes in pluripotent contexts.

Our analysis thus suggests that an ancestral LTR48B was inserted in the primate lineage and acquired enhancer activity in pluripotent cells. This enhancer likely recruited genes in spatial proximity under its transcriptional control, wiring these genes into the pluripotency network. In the hominoid lineage, the ERV insertions that gave rise to the *XACT* and *T113.3* genes hijacked the LTR48B ancestral enhancer. This raises the question as to whether this LTR-derived enhancer influences the expression of neighboring genes in species where *XACT* and *T113.3* are not present. Indeed, in rhesus macaque iPSCs, the protein-coding gene *HTR2C* is expressed at higher levels than its hominoid orthologs (Supplementary Fig. 1A). This suggests that the LTR48B enhancer could regulate the expression of the *HTR2C* gene in non-hominoid primates, and that the appearance of the *XACT* and *T113.3* genes insulated the *HTR2C* locus from the transcriptional control of the enhancer in hominoids.

Altogether, our data reveal the molecular mechanisms coupling the expression of the lncRNA *XACT* to human pluripotency, providing a working model for how its expression is linked to early embryonic stages and turned off during cellular differentiation. Interestingly, although *XACT* expression is dynamically regulated during human development, this is not the case of *XIST*, which is turned on soon after zygotic genome activation, stays active across pre-implantation stages, and, in females, remains expressed throughout post-implantation development and adulthood. Thus, human XCI is disconnected from the dynamics of *XIST* expression and rather seems to depend on the ability of *XIST* to initiate silencing. This is in contrast to mouse, in which XCI is mostly regulated by a dynamic and temporal control of *Xist* expression[26], with *Xist* being turned on upon differentiation. Several mechanisms involving the core pluripotency factors have been proposed to prevent the upregulation of *Xist* until differentiation takes place[56]. The mechanisms connecting human XCI to the developmental state are still largely unknown. Although the function of *XACT* has yet to be determined, it is tempting to speculate that the coupling of *XACT* expression to the pluripotent context, mediated by the LTR48B enhancer, could provide a mechanistic link between XCI and developmental timing. On a more general perspective, probing for TE-driven changes in conserved lncRNAs, such as *XIST*, or novel hominoid-specific lncRNAs, such as *XACT*, will allow a better understanding of the evolution of the XCI mechanism and what role TEs had in maintaining an essential function, while making it evolutionarily plastic.

## Methods

**Cell culture**. Experiments on hESCs have been approved by the Agence de la Biomedecine and informed consent was obtained from all subjects. The hESC lines (H1 and H9, obtained from the WiCell Research Institute, and WIBR2, obtained from the Whitehead Institute for Biomedical Research) were maintained on Matrigel-coated culture dishes (Corning) in mTeSR1 medium (Stem Cell Technologies) and were cultured in a 37 °C incubator with 20% $O_2$ and 5% $CO_2$. Cells were routinely passaged as clumps using gentle Cell Dissociation Reagent (Stemcell Technologies) approximately every 3–4 days. For experiments requiring single-cell suspension, cells were detached using Accutase (Stemcell Technologies) and plated on mTeSR1 supplemented with 10 μM of ROCK inhibitor (Y-27632, Merck Millipore). Differentiation experiments were carried out for 10 days on gelatin-coated 6-well plates by transitioning cells in Dulbecco's modified Eagle medium (ThermoFisher Scientific) with 10% of fetal bovine serum, 2 mM L-glutamine, and 0.1 mM non-essential amino acids (ThermoFisher Scientific).

**Knockdown using siRNA and LNA GapmeRs**. All transfections were performed using Lipofectamine RNAiMax (ThermoFisher Scientific), according to the manufacturer's recommendations. LNA GapmeRs used in this study were designed using the Exiqon online tool, with sequences spanning different regions of the XACT and T113.3 transcripts. For XACT and T113.3 KDs, cells were reverse transfected with LNA GapmeRs (Exiqon, QIAGEN) at a final concentration of 50 or 100 nM (with similar results). A scrambled LNA GapmeR and a LNA GapmeR targeting MALAT1, labeled with fluorescence amidite, were used as negative and transfection controls, respectively. Cells were collected at 24 or 48 h post transfection (with a stable efficiency of KD for both time points).

For KD of pluripotency factors, cells were transfected with 50 nM of siRNAs specific for POU5F1 (Ambion s10872), NANOG (Amibon s36650), or SOX2 (Dharmacon ON-TARGETplus SMARTpool L-011778–00–0005). siRNAs for MALAT1 (Ambion 4390843) and a scrambled siRNA (Ambion 4455877) were used as positive and negative controls, respectively. When co-transfecting multiple siRNAs, 25 nM of each siRNA was used. After transfection of $1 \times 10^6$ cells, these were plated in one well of of a six-well plate previously coated with Matrigel (Corning), with 10 μm of Y-2763 (Merck Millipore). Transfected cells were collected 72 h after transfection. Sequences of siRNAs are provided in Supplementary Table 1

**RNA-FISH on hESCs**. Cells grown on 12 mm Matrigel-coated coverslips (Corning) were fixed 24 to 48 h later in a 4% paraformaldehyde/phosphate-buffered saline solution (Electron Microscopy Science) for 12 min at room temperature. Cells were then permeabilized for 5 min in ice-cold cytoskeletal buffer (NaCl 100 mM, sucrose 300 mM, $MgCl_2$ 3 mM, PIPES 10 mM) supplemented with 0.5% Triton, 1 mM EGTA, and 2 mM Vanadyl Ribonucleoside complex (VRC, New England Biolabs), washed three times with 70% ice-cold ethanol, and kept at −20 °C.

Fluorescent probes were obtained by nick-translation, with either spectrum-green or spectrum-orange dUTPs (Abbott Molecular), as previously described[35]. The following probes were used in this study: XACT (RP11–35D3, BACPAC), XIST (a 10 kb fragment corresponding to XIST exon 1, gift from Dr C. Brown, University of British Columbia), and T113.3 (WI2–767I20, BACPAC) (Supplementary Table 2).

For hybridization, ~100 ng of labeled probes (enough for three 12 mm coverslips) were precipitated with 10 μg of sheared salmon sperm DNA (ThermoFisher Scientific) and 5 μg of human Cot-1 DNA (ThermoFisher Scientific), and then denatured in deionized formamide (Merck, Sigma-Aldrich) for 7 min at 75 °C. Denatured probes were then competed with the human Cot-1 DNA by incubating 30 min at 37 °C. Coverslips were dehydrated in sequential washes of 80%, 90%, and 100% ethanol just prior to overnight incubation with the probes at 37 °C in 50% formamide/50% hybridization buffer (4× SSC (saline-sodium citrate), 20% dextran sulfate, 2 mg/ml bovine serum albumin, and 2 mM VRC). After three 50% formadhyde/2× SSC washes and three 2× SSC washes at 42 °C for 5 min, coverslips were mounted in Vectashield containing 4′,6-diamidino-2-phenylindole (DAPI; Vector Laboratories).

**Microscopy and image analysis**. All images were acquired with a fluorescence DMI-6000 inverted microscope with a motorized stage (Leica) using a HCX PL APO ×100 oil objective and a CCD Camera HQ2 (Roper Scientifics) using the Metamorph software (version 7.04, Roper Scientifics). Approximately 40 optical Z-sections were collected at 0.5 μm steps across the nucleus for each wavelengths (DAPI [360 nm, 470 nm], fluorescein isothiocyanate [470 nm, 525 nm], and Cy3 [550 nm, 570 nm]). Stacks were processed using ImageJ. Throughout the manuscript, the three-dimensional FISH experiments are represented as a two-dimensional projection of the stacks (maximum intensity projection).

**RNA extraction and RT-qPCR**. Total RNAs were extracted from cells using TRIzol (ThermoFisher Scientific) and treated using the DNA free kit (ThermoFisher Scientific) for 1 h at 37 °C, to remove genomic DNA contamination. Five hundred nanograms of total RNAs were reverse-transcribed using SuperScript IV reverse transcriptase (ThermoFisher Scientific) and random primers (Promega). cDNA levels were measured using real-time qPCR with the Power SYBR Green PCR master mix (ThermoFisher Scientific) on a ViiA-7 real-time thermal cycler (Applied Biosystems). All samples were analyzed in technical duplicates. Normalization was performed using the reference gene GAPDH following the ΔCT or the ΔΔCT method. Sequences of RT-qPCR primers are provided in Supplementary Table 1.

**Characterization of the T113.3 transcript**. For the reconstruction of the T113.3 transcript, bam and bigWig files (GSM758573) were downloaded from the ENCODE website[57]. Transcripts were assembled using Scallop v.0.10.4[37] with default parameters. bigWig and corresponding Scallop reconstruction were visualized using the IGV genome browser. The T113.3 transcript was cloned from cDNA of H1 hESCs. PCR was performed using Platinum Taq DNA polymerase (ThermoFisher Scientific), using primers complementary to the first and last exons of T113.3 (primer sequences are provided in Supplementary Table 1). To allow the amplification of a bigger number of potential transcripts, an elongation time of 2 min 30 sec per cycle, was used. The PCR products were resolved on a 2% agarose gel, purified using a NucleoSpin Gel and PCR clean-up kit (Macherey-Nagel), cloned using the TOPO TA Cloning Kit (ThermoFisher Scientific) and sequenced.

**CRISPR/Cas9 deletions**. Single guide RNA (sgRNA) sequences flanking each target region were obtained using the web-based tool CRISPOR (http://crispor.tefor.net/) and can be found in the Supplementary Information. For each targeted region, upstream and downstream sgRNAs were cloned under an U6 promoter into the pSpCas9(BB)−2A-GFP (Addgene #48138) and the pSpCas9(BB)-2A-mCherry (generated in house, by replacing the green fluorescent protein (GFP) with a mCherry reporter using the NEBuilding HiFi DNA Assembly Cloning Kit [New England Biolabs]). All sgRNAs sequences can be found in the Supplementary Table 1 and vectors are listed in Supplementary Table 4.

Using the Amaxa 4D-NucleofectorTM system (Lonza), $1 \times 10^6$ H1 hESCs were transfected with 2.5 μg of both plasmids (to a total of 5 μg). Cells were sorted by fluorescence-activated cell sorting (INFLUX 500-BD BioSciences) 48 h after transfection. Double-positive cells (100, 200 or 400; GFP+/mCherry+) were plated into a matrigel or Laminin-coated 6 cm petri dish in mTeSR supplemented with 1× CloneR (Stemcell Technologies). Individual colonies were manually picked into 96 wells ~10 days after transfection. Deletions and inversion events were screened by PCR (primer sequences can be found in Supplementary Table 1).

**CRISPRi, lentiviral production, cell infection**. sgRNAs targeting either XACT or T113.3 promoters were designed using the web-based tool CRISPOR (http://crispor.tefor.net/) and cloned into the pLKO5.sgRNA.EFS.tGFP vector (Addgene #57823). sgRNA sequences can be found in Supplementary Table 1. Lentiviral particles were produced by transient transfection of HEK293T cells (ATCC® CRL-3216™) with the packaging plasmids pMD2.G (Addgene #12259) and psPAX2 (Addgene #12260), together with a lentiviral dCas9/KRAB-mCherry construct using calcium phosphate. Lentiviral particles carrying sgRNA-expressing vectors were also obtained from HEK293T cells using the same method. The culture media was collected 48 h after transfection of HEK293T and lentiviral particles were concentrated by ultracentrifugation. For each construct, the lentiviral titer was determined by infection of HEK293T cells and FACS analysis.

For CRISPRi, one million H1 hESCs were infected in suspension with lentiviral particles containing the dCas9-mCherry-KRAB (multiplicity of infection (MOI) of 10) and sorted by FACS (INFLUX 500-BD BioSciences) every week for 2–3 weeks, to enrich for mCherry-expressing cells (expressing dCas9-mCherry-KRAB). This cell line was then infected with lentiviral particles containing single sgRNA constructs (MOI of 10) targeting the TSS of either XACT or T113.3, and sorted as above, to enrich for GFP- and mCherry-expressing cells (co-expressing dCas9-mCherry-KRAB and sgRNAs). Cells infected with a lentiviral particle containing an empty sgRNA-expressing vector were used as control.

**Chromatin immunoprecipitation**. ChIP experiments were performed in H1 hESCs stably expressing the CRISPRi constructs, as previously described[58]. Briefly, cells were crosslinked with 1% formaldehyde for 10 min, then quenched with 0.125 mM glycine for 5 min. Cells were then incubated 30 min in swelling buffer (5 mM PIPES pH 8, 85 mM KCl, 0.5% NP-40). Chromatin was extracted and sonicated in TSE150 buffer (0.1% SDS, 1% Triton, 2 mM EDTA, 20 mM Tris-HCl pH 8, 150 mM NaCl, and protease inhibitors) with a Bioruptor Sonication System (Diagenode, UCD-200). Five micrograms of sonicated chromatin per sample was incubated overnight with 2–5 μg of antibody/protein A magnetic beads complexes (2 μl per IP of anti-H3K4me3 and anti-H3K27Ac; 2 μg per IP of anti-H3K9me3). The beads were subsequently washed in TSE150, TSE500 (20 mM Tris-HCl pH 8, 2 mM EDTA, 0.1% SDS, 1% Triton X-100, 500 mM NaCl), washing buffer (10 mM Tris-HCl pH 8, 1 mM EDTA, 250 mM LiCl, 0.5% NP-40, 0.5% Na-deoxycholate) and finally twice in TE buffer. Elution from the beads was done in TE/1% SDS. Samples were then reverse-crosslinked at 65 °C overnight. DNA was finally purified with phenol–chloroform and resuspended in water. The samples were analyzed by qPCR. Primer sequences are provided in Supplementary Table 1 and the list of antibodies in Supplementary Table 3.

**Protein extraction and western blotting**. Total proteins were extracted using RIPA buffer (50 mM Tris-HCl pH 8, 1 mM EDTA, 0.5 mM EGTA, 1% Triton X-100, 0.5% sodium deoxycholate, 0.1% SDS, 150 mM NaCl, and protease inhibitors) at 4 °C for 30 min. Extracts were sonicated on a Bioruptor Sonication System (Diagenode, UCD-200) for 10 min (30 s ON, 30 s OFF). Proteins were quantified using the BCA protein assay kit (ThermoFisher Scientific). Total protein extracts were resuspended in 5× Laemmli buffer at a final concentration of 500 μg/mL. Ten micrograms of denatured proteins were loaded into a 4–12% gradient polyacrylamide gel (ThermoFisher Scientific) and transferred onto Invitrolon polyvinylidene difluoride membranes (ThermoFisher Scientific). The membranes were blocked for 1 h with 5% milk in TBST (10 mM Tris pH 8.0, 150 mM NaCl, 0.5% Tween 20) and incubated overnight at 4 °C with the following antibodies: anti-NANOG (Abcam ab21624, 1:200), anti-SOX2 (Abcam ab97959, 1:1000), anti-OCT4 antibody (Abcam ab181557, 1:1000), anti-TUBULIN (Sigma-Aldrich T9026, 1:10 000), and anti-H3.3 (Merck Millipore 09–838, 1:1000) (Supplementary Table 3). A peroxidase-conjugated antibody was used to reveal the proteins of interest with the Pierce ECL Western blotting substrate (ThermoFisher Scientific). Uncropped scans of blots are provided in the Source Data file

**XACT/T113.3 sequence conservation and synteny**. XACT and T113.3 sequences were selected from the hg38 human genome assembly for comparison with orthologous sequences from various primate species. These orthologous sequences were identified using BLASTN[59] and confirmed with liftOver[60] on panTro5 (chimpanzee), gorGor4 (gorilla), nomLeu3 (gibbon), rheMac8 (rhesus macaque), and calJac3 (marmoset) genome assemblies. Synteny conservation for XACT, T113.3, and neighboring protein-coding genes (LHFPL1, AMOT, HTR2C and LRCH2) was analyzed for the above sequences.

Sequence identities between hg38 and other primates was determined using the MAFFT multiple alignment tool[61] with default parameters. Each primate sequence was independently aligned to hg38 orthologous sequence (using the above mentioned genome assemblies, with the exception of gorilla, in which we used gorGor5). Identity was calculated counting the number of nucleotide matches divided by multiple sequence alignment length. Due to incomplete assemblies of multiple primate species, nucleotides annotated as "N" and neighboring sequencing gaps were discarded and not considered for identity calculation. For the protein-coding genes, only the exons were taken into account to calculate the sequence conservation.

**Pluripotency factor-binding sites analysis**. To detect pluripotency transcription factors binding regions, Rsat matrix-scan[62] was performed using the XACT/T113.3 enhancer hg38 sequences and the following Position-Specific Scoring Matrices (PSSMs): the transfac M01307 for OCT4, M01123 for NANOG, and the jaspar MA0143.1 for SOX2. The most probable sequences (lowest p-values) were then selected as putative binding sites. The same method was used for each primate enhancer sequences. P-values were calculated using Rsat matrix-scan tool.

LTR48B human consensus sequences and logo were extracted from the Dfam website[63]. A multiple alignment with MAFFT was performed to compare the XACT/T113.3 enhancer sequence with the LTR48B consensus sequence. P-values were calculated as described above. In parallel, a Hidden Markov Model file was extracted from Dfam website. This file was converted to a PSSM using hmmlogo from hmmer tool (hmmer.org). From this matrix, the sequences corresponding to the identified binding sites in the XACT/T113.3 enhancer were selected. Consensus and reference matrices were compared using RSAT compare-matrices.

For ChIP-seq analysis, OCT4 and NANOG binding profiles on LTR48B and LTR7 were analyzed using available ChIP-seq datasets: SRR6177948 for OCT4, SRR6177944 for NANOG, and SRR6177932 for IgG[48]. LTR7 and LTR48B sequences in the human genome were selected from the RepeatMasker database[64]. ChIP-seq reads were aligned with hg38 using bowtie2 aligner[65] using the following parameters: "bowtie2 -p < threads > –end-to-end–no-mixed–no-discordant–minins 100–maxins 1000 -x hg38". This assigns reads with multiple hits of similar mapping quality to one of those locations randomly. To select only uniquely mapped reads, we used a previously reported custom script[66]. SAM files were sorted, compressed and indexed using samtools sort and samtools index[67]. PCR duplicates were then removed using MarkDuplicates from the Picard tool (http://broadinstitute.github.io/picard/). bigWigs were obtained with bamCompare from deepTools[68], which calculates the log2 ratio between the ChIP-seq and a ChIP-seq IgG control. A matrix for the relative-coverage of reads in the bigWig was then calculated for all the LTRs belonging to the LTR48B and LTR7 subfamilies using computeMatrix (deepTools). This matrix was finally plotted as a heatmap with plotHeatmap or as a binding profile around subfamilies of LTRs with plotProfile (deepTools).

To calculate the number of uniquely mapped reads aligning to different classes of LTRs, we calculated the number of reads that aligned to each individual LTR using samtools bedcov. For each individual LTR, a reads per kilobase per million mapped reads value was calculated. A control set of sequences was extracted taking 1000 random sequences from hg38 of a length of 390nt (average length of LTR48B and LTR7 elements in the genome). ggplot2 and ggpubr[69] were used to create boxplots and determine statistical values (t-test). Peak calling was performed with MACS2 with default parameters, using IgG ChIP-seq as a control sample. To

calculate the number of peaks on LTR7 and LTR48B, we used bedtools intersect to cross the LTRs in the genome to the called peaks.

**Reporting summary**. Further information on research design is available in the Nature Research Reporting Summary linked to this article.

## Data availability

All data generated or analyzed during this study are included in the published article, its Supplementary Information, and in the Source Data file, or from the corresponding authors upon reasonable request. A reporting summary for this article is available as a Supplementary Information file.

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

## Acknowledgements

We are grateful to members of the Rougeulle lab and members of the Epigenetics and Cell Fate Department for helpful discussion and evaluation of the work leading to this publication. We thank Anne-Valérie Gendrel, Céline Morey, Déborah Bourc'his, Sophie Polo, Miguel Branco, and Andrea Cerase for critical reading of the manuscript and valuable feedback. We also thank the Microscopy and Vectorology Platforms at the Epigenetic and Cell Fate Department, and the ImagoSeine cell-sorting facilities at the Institut Jacques Monod for access to instruments and technical advice. This study was supported by the LabEx "Who Am I?" (ANR-11-LABX-0071) and the Université de Paris IdEx (ANR-18-IDEX-0001) funded by the French Government through its "Investments for the Future" program. This work was also supported by the European Commission Network of Excellence EpiGeneSys (HEALTH-F4–2010–257082), Agence Nationale pour la Recherche (ANR-14-CE10–0017–01), and Ligue contre le cancer. M.C. has received financial support from the European Commission Network of Excellence EpiGeneSys (HEALTH-F4–2010–257082), Agence Nationale pour la Recherche (ANR-14-CE10–0017), and Labex Who Am I (ANR-11-LABX-0071). M.M. was supported by La Ligue contre le cancer. L.E.C. is supported by a fellowship from the French Ministry of Education and Research. O.R. was supported by a fellowship from the French Ministry of Education and Research, from the French Medical Research Foundation (FRM), and from the Labex Who Am I (ANR-11-LABX-0071).

## Author contributions

M.C. and C.R. conceived the project. M.C., M.M., and C.R. planned the experiments. M.C., M.M., L.E.C., C.H., J.S., T.M.L.A., and O.R. performed the experiments. M.C. and C.R wrote the manuscript. All authors commented on and revised the manuscript.

## Competing interests

The authors declare no competing interest.
