## [Peer Review File · Nature Communications]

Reviewers' comments:

Reviewer #1 (Remarks to the Author):

This manuscript presents an assessment of inter-regulatory control of a pair of adjacent X-linked lncRNA genes. Both are expressed only early in human ESC differentiation, and unlike XACT which forms a cloud the T113.3 transcript localizes in a small focal 'dot'. The manuscript very thoroughly explores the regulation of these genes, concluding that ERV integration has provided both promoter and enhancer for early developmental regulation. The control and exaptation of retroelements early in development has been of growing interest, and this manuscript presents a detailed exploration.

Biologically, the XACT gene has been proposed to be a 'competitor' for XIST accumulation; however, this manuscript does not pursue impact on XIST or function of the genes, only the impact of XACT and T113.3 on each other. While Figure 1 cites hg38 and UCSC browser (presumably, as the text legend states USCS), when I load either hg19 or hg38 in UCSC the search function does not find T113.3. The Lu et al. Scientific reports article (Lu Q, Hu Y, Sun J, Cheng Y, Cheung KH, Zhao H. A statistical framework to predict functional non-coding regions in the human genome through integrated analysis of annotation data. *Sci Rep.* 2015;5:10576. Published 2015 May 27. doi:10.1038/srep10576) suggests functionality for T113.3, and was not cited.

Beyond that, various transcripts as identified by ESTs are present in the region and not discussed.

Detailed comments on specific aspects of the manuscript follow.

Introduction:

The introduction is thorough and cites many reviews. While focused on the ERVs, there is also a discussion of X inactivation. The statement regarding timing of XIST expression seemed to be more definitive than the literature supports, given the rarity of tissues. It would be useful to include XIST expression in Figures 3-5 (perhaps only as supplemental) as was done for supplemental Figure 2, to demonstrate independence of XACT from XIST.

Results:

Figure 1: As noted above, I did not find T113.3 labelled in UCSC, but saw other ESTS in the region. It would be useful to address variants (both of transcripts and names). Furthermore, the region is remarkably repeat-rich and discussion of the conservation of repeat as well as the ones examined would be useful.

I would like to see more detail on the predicted 'origin' of XACT and T113.3 – perhaps a supplemental figure showing more repeats across the region in pre (e.g. Rhesus) and post (e.g. Gibbons)?

Figure 2: The transcripts are shown to have strong correlation with each other, and the figures are clearly labelled. Is there any correlation with HTR2C?

The apparent 'down-regulation' of HTR2C upon XACT/T113.3 integration (supp. Figure 1) is discussed later and is interesting. Does this also apply in somatic cells, or only iPSCs? If somatic cells, then this could be explored in more species. Also, with respect to the gain of the LTR48B enhancer in Figure 6 (although it seems likely all these activities are restricted to embryonic cells).

In the supplementary figures it appears the the ES line has stronger XACT expression than the embryonic cells. Is this true for multiple ES lines (there are biological replicates, but are those all H1, or a variety of ES?)?

Figure 3: The use of CRISPRi shows clear impact of T113.3 repression on XACT expression, but not vice-versa, with CRISPRi resulting in increased H3K9, but not spread to the other gene. The supplemental does show impact of T113.3 repression on H3K4me3 and H3K27ac at XACT, but also H3K27 (and possibly H3K4me3, the effect is very small) impact at T113.3 with sgXACT. This latter effect was interestingly not reflected in expression changes, and should be discussed. The nature of the biological (I presume) replicates should be discussed in the legend. Were each an independent infection and FACS sorting?

Figure 4: Both knockout and RNA knockout by LNA have no impact on the other gene. Here the consistent color scheme is very helpful for the reader.

Figure 5: Oregano seemed to identify SMARCA4 as a candidate for XACT regulation, but it was not mentioned (perhaps it is not expressed in embryonic cells?).

The text focusses on the H3K27ac peaks, I do not see much alignment of the peaks at the putative yellow enhancer. What are the peaks (upstream and downstream of the CTCF site, between the putative enhancer and the T113.3 start site)? Are these other repetitive elements? The concordance with the results in supplementary Figure 3 is also not clear to me with respect to location of peaks. Perhaps the primers used in S3 could be shown in Figure 5. I also don't see how this panel A matches Figure S5 panel A, where the single H3K27ac peak between the 2 promoters is more clear.

Both promoters also have combinations of OCT4, SOX2, NANOG, CTCF. Thus, the knock-down of these proteins by siRNA, could have functions through the promoter as well as the enhancer. The fact that individually the only effect is upregulation of T113.3 with siOCT4, yet in combination SOX2 and OCT4 siRNA abrogates expression of both genes suggests more than a simple regulatory action.

Reviewer #2 (Remarks to the Author):

Casanova and Rouguelle and colleagues present a study of the mechanisms governing the expression of the XACT lncRNA during early development and in the context of X-chromosome inactivation. They use a combination of perturbation techniques to investigate the potential roles of neighbouring lncRNA, repetitive elements and enhancers. The outcome is the demonstration that a relatively recent, transposable element derived enhancer controls XACT activation during preimplantation development, is activated by the core pluripotency transcription factors, and thereby may sit atop the regulatory cascade driving dosage compensation in human and other Great Apes.

Although the overall impact of these findings may not be of the highest order, nevertheless one must appreciate the care and rigour of this work in meticulously isolating the regulatory mechanism for XACT. The usage of CRISPR technologies here is top rate. Overall the science is of highest quality, a hallmark of this research team for which they justifiably have an excellent reputation.

Overall I have studied the paper carefully and find little to criticize or query from either a technical or theoretical point of view. My comments here are relatively minor.

L60 On the subject of the "alliance" between lncRNAs and TEs, this topic was proposed in a review article several years ago and might be appropriate to cite here (Johnson, Guigo 2014)

L72 A good citation for the contribution of TEs in XIST is from Elisaphenko and colleagues (2008), it is perhaps better to cite primary literature here rather than review articles.

L140 There is no citation or figure for this statement about XACT in chimpanzee.

T113.3 – I felt that this gene was somewhat neglected in the paper. Was it previously studied? What is its protein-coding status? Where is its subcellular localization? Does it have any function? To what extent is it conserved. Perhaps this could be expanded on a little at an appropriate point in the manuscript.

TEs in mature lncRNAs: The intersection of TEs and lncRNAs can be classified in two ways: as regulators of lncRNA transcription, or in the context of mature lncRNA transcript. It would be interesting if the authors comment on the latter, since apparently both XACT and T113.3 transcripts also contain TE fragments. These have been linked to several roles, including protein binding (eg in XIST), DNA binding (Fendrr), and nuclear localization (Lubelsky, Carlevaro). Again, this might be mentioned, if the authors find it appropriate.

Outcome of KO: Perhaps I missed it, but does knockout of either XACT or T113.3 (either DNA or RNA) give rise to any detectable effect on either differentiation or X-Chromosome inactivation in ES cells?

Reviewer #3 (Remarks to the Author):

In the manuscript by Casanova et al., the authors explore the regulation of XACT and the nearby T113.3. They identify an LTR48B class of TE that is bound by pluripotency regulators and acts as an enhancer for XACT and T113.3. The authors go on to delete the LTR locus, and confirm its activity as a bonafide enhancer of XACT and T113.3. This is one of the few studies that will contribute to the functionalising of TEs as enhancer elements.

The work is well performed, comprehensive, and supports the authors conclusions. I just have a few minor comments that the authors might like to address.

Minor points:

Line 76-79: I think this study does not really contribute to the discussion of the timing of activation, which is only briefly explored using the single cell embryo data. I suggest the author rephrase or delete this sentence.

Line 143: The authors say that XACT and T133.3 lack promoters in macaque, but it is unclear how Supp Figure 1 shows this, as it just shows the expression of these genes and may reflect differences in the iPSCs, rather than in the XACT and T133.3 expression. IT might be useful to briefly summarize the X chromosome status in the different species iPSC lines and the expression of XACT and T133.3.

Figure 2C: These figures are not the best, and the correlations are not strong. Can this analysis be performed using more recent higher quality sc-RNA-seq? The Yan et al., data is not the best quality. Additions of bulk RNA-seq data may also be useful to simplify the message in these panels.

Figure 2A and SuppFigure 2A: The schematic views next to the FISH are intended to help the reader. But in their current form they are nearly unintelligible. In Supp 2A the grey thing is the X chromosome? In 2A, what is the dotted line?

Figure 6C: The motif presented as 'SOX' is actually the compound SOX-OCT motif. SOX would bind to

the CATTAT sequence, and the following TTGCAG sequence I think would not be capable of recruiting OCT4. These distinctions do not impact on the conclusions, but should be noted.

Reviewer #1 (Remarks to the Author):

This manuscript presents an assessment of inter-regulatory control of a pair of adjacent X-linked lncRNA genes. Both are expressed only early in human ESC differentiation, and unlike XACT which forms a cloud the T113.3 transcript localizes in a small focal 'dot'. The manuscript very thoroughly explores the regulation of these genes, concluding that ERV integration has provided both promoter and enhancer for early developmental regulation. The control and adaptation of retroelements early in development has been of growing interest, and this manuscript presents a detailed exploration.

Biologically, the XACT gene has been proposed to be a 'competitor' for XIST accumulation; however, this manuscript does not pursue impact on XIST or function of the genes, only the impact of XACT and T113.3 on each other. While Figure 1 cites hg38 and UCSC browser (presumably, as the text legend states USCS), when I load either hg19 or hg38 in UCSC the search function does not find T113.3.

- "USCS" on the legend was a typo and was corrected on the revised version of the manuscript (line 837).

- The *T113.3* transcript was identified in a previous study from our group (Vallot *et al.*, 2013), as being produced by a transcription unit located 5' of *XACT*, expressed from the plus strand, that we named *T113.3* based on its genomic coordinates. This transcript was characterized as a spliced, mostly cytoplasmic transcript and with a TSS mapped by 5' RACE to a peak of H3K4me3 located 48 kb upstream of the TSS of *XACT*. The *T113.3* gene was therefore not annotated previously (and, in fact, the transcripts that are currently annotated as *XACT* in the current GENCODE / Ensembl annotation are not the transcripts we described in our 2013 Nature Genetics paper). We have started the procedure to properly annotate both genes and respective hESCs transcripts in Gencode and Genbank.

The Lu et al. Scientific reports article (Lu Q, Hu Y, Sun J, Cheng Y, Cheung KH, Zhao H. A statistical framework to predict functional non-coding regions in the human genome through integrated analysis of annotation data. Sci Rep. 2015;5:10576. Published 2015 May 27. doi:10.1038/srep10576) suggests functionality for T113.3, and was not cited.

This is indeed an important reference that we have used in the past, but did not think to include in our manuscript. Although the message of this manuscript does not focus on the role of either *XACT* or *T113.3*, we do feel that reviewer 1 and reviewer 2 raised some issues regarding the information about the *T113.3* gene. We have thus added this reference to the manuscript (line 105), and extended the section on *T113.3* to include a more thorough description and characterization of this gene (line 100-105). We also provide a new panel (which appears in the revised manuscript as Sup. Fig. 1A, and which is provided below as Rebuttal Figure 1), which includes (i) H1 RNA-seq data, (ii) scallop transcript assembly to reconstruct the transcripts arising from *T113.3* in hESCs, (iii) results of cDNA cloning experiments in H1 hESCs and (iv) *in silico* analysis of the coding potential of this transcript, which suggests it is a lncRNA. A new section has been added to the M&M describing the characterization of the *T113.3* transcript in hESCs (lines 460-471).

A

Rebuttal Figure 1

Beyond that, various transcripts as identified by ESTs are present in the region and not discussed.

The reviewer is indeed correct in stating that many transcripts and ESTs are annotated in this region. However, *XACT* and *T113.3* are the two major transcripts found in this region in hESCs and therefore the focus of our study. This can be observed in the following figure containing CAGE and RNA-seq data from H1 hESCs that we prepared for the reviewer as Rebuttal Figure 2.

Rebuttal Figure 2

Detailed comments on specific aspects of the manuscript follow.

Introduction:

The introduction is thorough and cites many reviews. While focused on the ERVs, there is also a discussion of X inactivation. The statement regarding timing of XIST expression seemed to be more definitive than the literature supports, given the rarity of tissues. It would be useful to include XIST expression in Figures 3-5 (perhaps only as supplemental) as was done for supplemental Figure 2, to demonstrate independence of XACT from XIST.

We agree with the Reviewer's comment regarding the timing of XCI in human embryos. In order to address this, we have now changed the text to focus on the uncoupling between *XIST* expression and inactivation, which has been properly documented (line 69-73). We will leave aside the issue of timing of silencing as this is still controversial, with very few studies tackling this question in a systematic manner.

Regarding the point raised about *XIST* expression in figures 3-5, we should recall that all of our functional studies were made in male H1 hESCs, where *XIST* is not expressed. As the dynamics of *XACT* and *T113.3* are similar in male and female hESCs, we decided to tackle our question about the transcriptional regulation of *XACT* in a cellular context that did not have the confounding effect of X inactivation. This rendered our functional assays, namely CRISPR KO and CRISPR interference, easier to accomplish and cleaner to interpret.

Finally, the question of *XACT* and *XIST* independence/interdependence is, obviously, a question that we are actively pursuing in our group and one that, we feel, belongs in a different set of studies.

[Redacted]

Results:

Figure 1: As noted above, I did not find T113.3 labelled in UCSC, but saw other ESTS in the region. It would be useful to address variants (both of transcripts and names). Furthermore, the region is remarkably repeat-rich and discussion of the conservation of repeat as well as the ones examined would be useful.

I would like to see more detail on the predicted 'origin' of *XACT* and T113.3 – perhaps a supplemental figure showing more repeats across the region in pre (e.g. Rhesus) and post (e.g. Gibbons)?

The repeat composition of this region has been previously described in a study from our group (Vallot et al, 2013). However, we have prepared some figures (Rebuttal Figure 4 to 6) in order to address these comments.

As we mention in our manuscript, the “origin” of the *XACT* and *T113.3* genes dates back to the common ancestor between Rhesus and Gibbon, when a series of ERVs were introduced in the *XACT/T113.3* locus, creating the promoters driving the expression of both genes. This is shown in Figures 1B and 1D of the manuscript and mentioned in the main text, as depicted below in Rebuttal Figure 4 (with both panels fused).

Rebuttal Figure 4

In regards to the evolution of the locus, *per se*, we do not believe that its repeat composition changed dramatically in the primate lineages (at least, not since the common progenitor of Rhesus and Gibbon). To corroborate this, we provide the reviewer with Rebuttal Figure 5 showing the conservation of the whole *XACT / T113.3* locus in primates, where it is evident that the locus is relatively well conserved.

Rebuttal Figure 5

Moreover, we analyzed the repeat composition of the locus in Human, Chimpanzee and Rhesus macaque and we provide Rebuttal Figure 6 to the Reviewer. The global repeat content of the whole locus is quite comparable between species. In addition, the nature and percentage of different classes of repeats has been kept stable across the different primate species analyzed. We hope that with all this information, we can convince the Reviewer about the “origin” of the two genes.

Rebuttal Figure 6

Figure 2: The transcripts are shown to have strong correlation with each other, and the figures are clearly labelled. Is there any correlation with HTR2C?

To address this comment, we provide Rebuttal Figure 7 for the Reviewer including qRT-PCR data for *HTR2C* expression during differentiation of WT and H1 *T113.3 KO* hESCs. Unlike *XACT* and *T113.3*, the expression of *HTR2C* is immediately shutdown upon differentiation. Thus, the expression dynamics of these genes is not correlated, at least during hESC differentiation.

Rebuttal Figure 7

The apparent ‘down-regulation’ of *HTR2C* upon *XACT*/*T113.3* integration (supp. Figure 1) is discussed later and is interesting. Does this also apply in somatic cells, or only iPSCs? If somatic cells, then this could be explored in more species. Also, with respect to the gain of the LTR48B enhancer in Figure 6 (although it seems likely all these activities are restricted to embryonic cells).

We do agree that the potential exclusion of the *HTR2C* gene from the transcriptional control of the LTR48B enhancer by the introduction of *T113.3* (or *XACT*) is an interesting concept. Analysis of *HTR2C* expression in multiple human and macaque somatic tissues from the Expression Atlas (<https://www.ebi.ac.uk/gxa/home>; with human data originating from Human protein Atlas, and Rhesus data from Merkin *et al.* Science 2012) reveals similar patterns of expression in both species, with *HTR2C* being silent in most tissues except brain (Rebuttal Figure 8). This argues against a differential impact of the LTR48B enhancer on *HTR2C* in somatic tissues between species with or without *T113.3* and *XACT*. Corroborating this conclusion, no expression difference is seen for *HTR2C* between WT and *T113.3* KO differentiated human cells derived from hESCs (Rebuttal Figure 7, days 3 to 10 of differentiation). *HTR2C* expression appears to be only transiently derepressed in KO cells at the onset of differentiation (Rebuttal Figure 7, day 2), which requires further investigation, but we feel is beyond the scope of this manuscript

Rebuttal Figure 8

In undifferentiated hESCs

(in which the LRR48B enhancer is active), the expression of *HTR2C* is slightly increased, but not to a point that reaches statistical significance (see Rebuttal Figure 7 and Sup. Fig. 4C). Whereas this suggests that in a pluripotent context, the expression of *HTR2C* in hESCs might be influenced by the LTR48B enhancer, in differentiated contexts, alternative mechanisms of transcriptional regulation overrule the influence that the enhancer could play on *HTR2C*.

In the supplementary figures it appears the the ES line has stronger *XACT* expression than the embryonic cells. Is this true for multiple ES lines (there are biological replicates, but are those all H1, or a variety of ES?)?

We should highlight that in the supplementary figure, as it is described in the legend (and main text), the cell line depicted is a female hESC line, WIBR2. We provide below Rebuttal Figure 9 showing that *XACT* and *T113.3* display different expression levels in various hESC lines (male H1 and female H9 and WIBR2 cell lines). We are not sure what accounts for these differences. One possibility is that this could be linked to the number of alleles (1 or 2) for the genes.

Rebuttal Figure 9

Figure 3: The use of CRISPRi shows clear impact of *T113.3* repression on *XACT* expression, but not vice-versa, with CRISPRi resulting in increased H3K9, but not spread to the other gene. The supplemental does show impact of *T113.3* repression on H3K4me3 and H3K27ac at *XACT*, but also H3K27 (and possibly H3K4me3, the effect is very small) impact at *T113.3* with sg*XACT*. This latter effect was interestingly not reflected in expression changes, and should be discussed. The nature of the biological (I presume) replicates should be discussed in the legend. Were each an independent infection and FACS sorting?

There is indeed a slight decrease in H3K27Ac upstream of *T113.3* (corresponding to the enhancer element) upon *XACT* CRISPRi, that is not reflected in *T113.3* expression change. The reason for this is unclear, but we can postulate that this subtle modification of H3K27 acetylation is not sufficient to alter the activity of the enhancer. In the absence of explanation, we feel that discussing this point will not add much to the manuscript. Nevertheless, we have removed the part of the data analysis in the results

section, which stated that these changes were “likely reflecting the transcriptional changes induced by the CRISPRi machinery” (line 185-186).

Regarding the nature of the replicates, we started by creating a cell line constitutively expressing dCas9-KRAB-mCherry. This cell line was then infected with lentivectors expressing different guides to target the regions of interest in the genome (the TSS of *XACT* and *T113.3*). We amended our material and methods to reflect this (lines 502-507).

Furthermore, we did two independent infections with the sgRNA constructs. In a first infection, we used four guides per site (infected individually). Two weeks after the infection, we analyzed the knockdown efficiency by qRT-PCR and chose the two best guides for each site (sgi285 and 286 for *T113.3* and sgi330 and 332 for *XACT*). We provide Rebuttal Figure 10 with results of this first experiment for the reviewer.

Rebuttal Figure 10

We confirmed that the knockdown was stable and decided to make a second round of infections. All the experiments presented in the manuscript are from this second round of infections. In the second experiment, we infected the cell line stably expressing dCas9-KRAB with the lentiviral vectors expressing the most efficient guides (individually). We collected cells every week, for four weeks, to measure the dynamics of knockdown. We provide Rebuttal Figure 11 representing this experiment.

In summary, the qRT-PCR data on the manuscript correspond to, at least, four different passages of hESCs infected with guides targeting either *XACT* or *T113.3*. For the ChIP, we collected chromatin from three different passages of hESCs expressing the different guides for each gene and pooled the data for the two guides and three independent passages.

Rebuttal Figure 11

Figure 4: Both knockout and RNA knockout by LNA have no impact on the other gene. Here the consistent color scheme is very helpful for the reader.

Figure 5: Oregon seemed to identify SMARCA4 as a candidate for XACT regulation, but it was not mentioned (perhaps it is not expressed in embryonic cells?).

In this study, we focus on the surrounding genomic regions that might be regulating the locus, which led us to identify the LTR48B enhancer. Using the CISTRROME database, we did identify binding of several “canonical” TFs around the regulatory regions of *XACT*, *T113.3* and the enhancer, such as *TEAD4* and *CEBPB*. We used siRNAs for *TEAD4* and *CEBPB* in H1 hESCs, but did not observe any downregulation of either *XACT* or *T113.3* (data not shown). Furthermore, *SMARCA4* does indeed bind the regulatory regions of the *XACT/T113.3* locus, as many other TFs. Nevertheless, these are not pluripotent specific-TFs and are ubiquitously expressed across many different tissues. We decided to focus our study only on pluripotent specific factors that bind within this region, justifying hence the shortlist of TFs we have tested (*OCT4*, *NANOG* and *SOX2*).

The text focusses on the H3K27ac peaks, I do not see much alignment of the peaks at the putative yellow enhancer. What are the peaks (upstream and downstream of the CTCF site, between the putative enhancer and the T113.3 start site)? Are these other repetitive elements? The concordance with the results in supplementary Figure 3 is also not clear to me with respect to location of peaks. Perhaps the primers used in S3 could be shown in Figure 5. I also don't see how this panel A matches Figure S5 panel A, where the single H3K27ac peak between the 2 promoters is more clear.

In our text, we do not mention peaks for H3K27ac, but rather a broad domain spanning the LTR48B region (lines 228). The resolution of the panel in Figure 5A, which covers just a few kbs, is very different from the panel in Fig S5A, which shows a region of a few hundred kbs. Nonetheless, the information contained in both panels is identical (it is exactly the same tracks that are displayed at different resolutions).

Both promoters also have combinations of OCT4, SOX2, NANOG, CTCF. Thus, the knock-down of these proteins by siRNA, could have functions through the promoter as well as the enhancer. The fact that individually the only effect is upregulation of T113.3 with siOCT4, yet in combination SOX2 and OCT4 siRNA abrogates expression of both genes suggests more than a simple regulatory action.

We thank the reviewer for this comment and have now changed our text to tone down the conclusions we made from the siRNAs experiments, saying that we cannot exclude that the results obtained could also be due to a direct influence of these TFs at the TSS of the genes (line 258-261). Nevertheless, if we consider the LTR48B CRISPR-mediated deletion, it seems that the binding of these TFs at the LTR48B enhancer is essential for the expression of both *XACT* and *T113.3*.

Reviewer #2 (Remarks to the Author):

Casanova and Rouguelle and colleagues present a study of the mechanisms governing the expression of the XACT lncRNA during early development and in the context of X-chromosome inactivation. They use a combination of perturbation techniques to investigate the potential roles of neighbouring lncRNA, repetitive elements and enhancers. The outcome is the demonstration that a relatively recent, transposable element derived enhancer controls XACT activation during preimplantation development, is activated by the core pluripotency transcription factors, and thereby may sit atop the regulatory cascade driving dosage compensation in human and other Great Apes.

Although the overall impact of these findings may not be of the highest order, nevertheless one must appreciate the care and rigour of this work in meticulously isolating the regulatory mechanism for XACT. The usage of CRISPR technologies here is top rate. Overall the science is of highest quality, a hallmark of this research team for which they justifiably have an excellent reputation.

Overall I have studied the paper carefully and find little to criticize or query from either a technical or theoretical point of view. My comments here are relatively minor.

We would like to acknowledge the Reviewer for the positive comments on our manuscript.

L60 On the subject of the “alliance” between lncRNAs and TEs, this topic was proposed in a review article several years ago and might be appropriate to cite here (Johnson, Guigo 2014).

This reference has now been added in the introduction section of the manuscript and we do agree it is an important and enticing concept to introduce our work (line 54-56).

L72 A good citation for the contribution of TEs in XIST is from Elisaphenko and colleagues (2008), it is perhaps better to cite primary literature here rather than review articles.

We have now added this reference to the introduction together with another important reference: Duret and colleagues (2006) (line 67).

L140 There is no citation or figure for this statement about XACT in chimpanzee.

The information for *XACT* expression in chimpanzee is presented in Sup. Fig. 1B. This citation is indeed missing from the text and has now been added (line 124).

T113.3 – I felt that this gene was somewhat neglected in the paper. Was it previously studied? What is its protein-coding status? Where is its subcellular localization? Does it have any function? To what extent is it conserved. Perhaps this could be expanded on a little at an appropriate point in the manuscript.

This is indeed an important point that was also raised by Reviewer 1. We changed the text to expand the information about *T113.3* (lines 99-105), by better referencing the previous characterization of the *T113.3* transcript made by Vallot et al, 2013. We have also added a reference to Lu *et al.* (Lu Q, Hu Y, Sun J, Cheng Y, Cheung KH, Zhao H. A statistical framework to predict functional non-coding regions in the human genome through integrated analysis of annotation data. *Sci Rep.* 2015;5:10576. Published 2015 May 27. doi:10.1038/srep10576), which suggests functionality for *T113.3*. Furthermore, we provide a new panel (Sup. Fig. 1A of the revised manuscript, also shown here as Rebuttal Figure 1) containing RNA-seq tracks for *T113.3* in H1 hESCs and scallop transcript assembly for the same datasets. In addition, we provide a scheme of our own cDNA cloning and sequencing and of the coding potential of the respective transcript. We hope that with these changes, we can contextualize the *T113.3* gene better.

TEs in mature lncRNAs: The intersection of TEs and lncRNAs can be classified in two ways: as regulators of lncRNA transcription, or in the context of mature lncRNA transcript. It would be interesting if the authors comment on the latter, since apparently both *XACT* and *T113.3* transcripts also contain TE fragments. These have been linked to several roles, including protein binding (eg in *XIST*), DNA binding (Fendrr), and nuclear localization (Lubelsky, Carlevaro). Again, this might be mentioned, if the authors find it appropriate.

This is indeed a very good and exciting point. It couples extraordinarily well with the point previously made by the Reviewer about how TEs constitute functional blocks in lncRNAs. We find though, that this concept is better suited for an article exploring the role of the *XACT* lncRNA, which we do not do in the present manuscript. We will undoubtedly discuss this point in our future work exploring the role of *XACT*.

Outcome of KO: Perhaps I missed it, but does knockout of either *XACT* or *T113.3* (either DNA or RNA) give rise to any detectable effect on either differentiation or X-Chromosome inactivation in ES cells?

The results presented on Sup. Fig. 4C show that the deletion (or inversion) of the *T113.3* gene has no impact in the expression of pluripotency or lineage specific markers in hESCs. Moreover, H1 hESCs lacking *T113.3* show normal differentiation dynamics (Sup. Fig. 4D). In addition, we provide Rebuttal Figure 12 exemplifying the morphologies of WT, KO and INV *T113.3* H1 hESCs at different days of differentiation, where no obvious differences can be observed between clones.

Finally, we could not analyze the impact of *T113.3* KO in XCI, as all mutant hESCs have been made in male H1 hESCs. This is however, a question that we are actively pursuing in the lab.

Rebuttal Figure 12

Reviewer #3 (Remarks to the Author):

In the manuscript by Casanova et al., the authors explore the regulation of XACT and the nearby T113.3. They identify an LTR48B class of TE that is bound by pluripotency regulators and acts as an enhancer for XACT and T113.3. The authors go on to delete the LTR locus, and confirm its activity as a bonafide enhancer of XACT and T113.3. This is one of the few studies that will contribute to the functionalising of TEs as enhancer elements.

The work is well performed, comprehensive, and supports the authors conclusions. I just have a few minor comments that the authors might like to address.

Again, we would like to acknowledge the Reviewer for the positive comments.

Minor points:

Line 76-79: I think this study does not really contribute to the discussion of the timing of activation, which is only briefly explored using the single cell embryo data. I suggest the author rephrase or delete this sentence.

This is a point that was also raised by Reviewer 1. We therefore changed our text to put more emphasis on the decoupling between *XIST* expression and silencing and not on the timing of XCI in humans (lines 69-73).

Line 143: The authors say that XACT and T133.3 lack promoters in macaque, but it is unclear how Supp Figure 1 shows this, as it just shows the expression of these genes and may reflect differences in the iPSCs, rather than in the XACT and T133.3 expression. IT might be useful to briefly summarize the X chromosome status in the different species iPSC lines and the expression of XACT and T133.3.

In Fig. 1B, we show that Rhesus macaque lacks the promoter regions of both *XACT* and *T113.3*. This is mentioned on a sentence on the previous paragraph “Notably, the sequences corresponding to the promoter region of *XACT* and *T113.3* are conserved in hominoids, but not in rhesus macaque or more distant primate species (Fig. 1B).”.

The XCI status in closely related primate species (Rhesus macaque, Chimpanzee) is currently being investigated in the lab as part of a project that has recently been funded. From the data we have already gathered, primate iPSCs are similar to human pluripotent cells and they have all undergone XCI. This is although, sensitive information that will be made available as part of the study above and cannot be provided at the moment.

Figure 2C: These figures are not the best, and the correlations are not strong. Can this analysis be performed using more recent higher quality sc-RNA-seq? The Yan et al., data is not the best quality. Additions of bulk RNA-seq data may also be useful to simplify the message in these panels.

The datasets used for the correlation in Fig. 2C are from Petropoulos *et al.* (2016), which are the most recent single-cell pre-implantation datasets available. The correlations we obtained are high,

considering the technology used and the biological materials analyzed. The “older datasets”, correspond to the combined data from Blakeley et al. 2015, Xue et al. 2013 and Yan et al. 2013. This combined dataset was only used for the histograms displaying the expression dynamics of *XACT* and *T113.3* in Sup. Fig. 2C, as these embryos were sorted by developmental stage (whereas in Petropoulos, they were sorted by day of *in vitro* culture).

Finally, we do not think that bulk RNA-seq analysis would provide a better interpretation of the results obtained from the analysis of the scRNA-seq.

Figure 2A and SuppFigure 2A: The schematic views next to the FISH are intended to help the reader. But in their current form they are nearly unintelligible. In Supp 2A the grey thing is the X chromosome? In 2A, what is the dotted line?

We apologize for the lack of clarity of the schemes and confusion for the reviewer. In Sup. Fig. 2A, the “grey thing” is indeed the inactive X chromosome and the dotted line is the active X. We have remade the schemes in Fig. 2A and Sup. Fig. 2A hoping that they now help the reader better understanding the data.

Figure 6C: The motif presented as ‘SOX’ is actually the compound SOX-OCT motif. SOX would bind to the CATTAT sequence, and the ‘following TTGCAG sequence I think would not be capable of recruiting OCT4. These distinctions do not impact on the conclusions, but should be noted.

The Reviewer is totally correct. In fact, the matrix used for Figure 6C is MA0143.1, which is a SOX2 matrix (likely, with a degenerated/weak binding site for OCT4). We swapped matrixes (from the SOX-OCT MA0142.1 matrix to the SOX2 matrix, MA0143.1) during the elaboration of our manuscript and forgot to update the matrix name. This has now been corrected in the figures, figure legends and M&Ms (lines 567 and 936).

REVIEWERS' COMMENTS:

Reviewer #1 (Remarks to the Author):

I thank the authors for the detailed response to reviews. The inclusion of panel s1A is helpful in addressing the transcripts expressed in the current absence of their presence on UCSC. S1B shows an additional transcription peak on the + strand between T113.3 and HTR2C. A comment on this would be helpful - is it another lncRNA, and is it repeat-derived? The authors are (and were originally) clear that their knockdown experiments are done in male H1 cells, so my questioning XIST is unaddressable.

With regards to the replication, I believe that the statement of 'at least [three or four] different passages of hESCs infected with the noted guides' in the legend would be more clear to readers than only the addition to the methods (which I did not find as clear as the response to reviews).

Reviewer #2 (Remarks to the Author):

I thank the authors for addressing my comments comprehensively.

Reviewer #3 (Remarks to the Author):

The authors have addressed all of my comments

Reviewer #1 (Remarks to the Author):

I thank the authors for the detailed response to reviews. The inclusion of panel s1A is helpful in addressing the transcripts expressed in the current absence of their presence on UCSC. S1B shows an additional transcription peak on the + strand between T113.3 and HTR2C. A comment on this would be helpful - is it another lncRNA, and is it repeat-derived? The authors are (and were originally) clear that their knockdown experiments are done in male H1 cells, so my questioning XIST is unaddressable.

We thank the Reviewer for highlighting this transcription peak on the + strand between T113.3 and HTR2C. Scallop transcript assembly to reconstruct the transcript reveals essentially two isoforms, the major one being unspliced, and the minor one being made of 2 exons (see accompanying figure below). None of these isoforms displays coding potential using CPAT. This locus is fully derived from repeat elements, mainly HERVH, and appears poorly conserved (see figure below). A short comment has been added to the legend of Supplementary Figure 1.

With regards to the replication, I believe that the statement of 'at least [three or four] different passages of hESCs infected with the noted guides' in the legend would be more clear to readers than only the addition to the methods (which I did not find as clear as the response to reviews).

We have modified the legend to Figure 3 to clarify this.

Reviewer #2 (Remarks to the Author):

I thank the authors for addressing my comments comprehensively.

Reviewer #3 (Remarks to the Author):

The authors have addressed all of my comments